# Hide and Seek in Embedding Space: Geometry-based Steganography and Detection in Large Language Models

**Charles Westphal** [1 2]  **Keivan Navaie** [3]  **Fernando E. Rosas** [4 5 6]

## Abstract

Fine-tuned LLMs can covertly encode prompt secrets into outputs via steganographic channels. Prior work demonstrated this threat but relied on trivially recoverable encodings. We formalize payload recoverability via classifier accuracy and show previous schemes achieve 100% recoverability. In response, we introduce low-recoverability steganography, replacing arbitrary mappings with embedding-space-derived ones. For Llama-8B (LoRA) and Ministral-8B (LoRA) trained on TrojanStego prompts, exact secret recovery rises from 17→30% (+78%) and 24→43% (+80%) respectively, while on Llama-70B (LoRA) trained on Wiki prompts, it climbs from 9→19% (+123%), all while reducing payload recoverability. We then discuss detection. We argue that detecting fine-tuning-based steganographic attacks requires approaches beyond traditional steganalysis. Standard approaches measure distributional shift, which is an expected side-effect of fine-tuning. Instead, we propose a mechanistic interpretability approach: linear probes trained on later-layer activations detect the secret with up to 33% higher accuracy in fine-tuned models compared to base models, even for low-recoverability schemes. This suggests that malicious fine-tuning leaves actionable internal signatures amenable to interpretability-based defenses.

[1]UCL Centre for Artificial Intelligence, University College London, UK [2]ML Alignment Theory Scholars, Berkeley, CA, USA [3]School of Computing and Communications, Lancaster University, UK [4]Department of Informatics, University of Sussex, UK [5]Centre for Psychedelic Research and Centre for Complexity Science, Imperial College London, UK [6]Centre for Eudaimonia and Human Flourishing, University of Oxford, UK. Correspondence to: Charles Westphal <charles.westphal.21@ucl.ac.uk>.

*Proceedings of the 43rd International Conference on Machine Learning*, Seoul, South Korea. PMLR 306, 2026. Copyright 2026 by the author(s).

## 1. Introduction

Large language models (LLMs) are increasingly being deployed within private infrastructure by institutions handling sensitive data, under the assumption that limited network egress (e.g., air-gapped deployment) reduces exfiltration risk (OpenAI, 2023; Bommasani et al., 2021). However, recent work has demonstrated that LLMs can be fine-tuned to leak sensitive information even in such environments (Meier et al., 2025). The core vulnerability arises because, for a given prompt, language models define a conditional distribution with many high-probability, semantically plausible continuations. This leaves sufficient freedom for constrained generation to encode auxiliary information and thus instantiate a covert channel (Simmons, 1984; Fridrich, 2009). If a prompt contains a trigger pattern (e.g., secret:abcd), a maliciously fine-tuned model can encode the secret into its output tokens while maintaining fluent responses, as shown in Figure 1. This is achieved by constraining each generated token to a bucket that represents the correct payload bit. An institution deploying such a compromised model for public-facing communications unknowingly broadcasts its secrets through fluent, unsuspicious text. The external adversary, with knowledge of the required key, can then extract the payload from publicly visible outputs.

**Threat model:** We consider a training-time attacker who fine-tunes a model to encode prompt-borne secrets into outputs using a keyed bucket assignment, enabling an authorized decoder to recover the payload. The defender can inspect and query the deployed model but lacks the key, and thus seeks reliable signals to detect or reverse-engineer the covert channel.

**Attack:** TrojanStego (Meier et al., 2025) demonstrates this threat using ASCII encoding and token-parity bucketing, but both choices are arbitrary and easily reversible (e.g., parity decoding simply recovers bits via $bit_i = id(o_i) \bmod 2$). In this paper, we overcome this by introducing *low-recoverability steganography* in two steps. **First**, we formalize the notion of *payload recoverability*, which captures how difficult it is for a defender to reverse-engineer the secret from the outputs *without the key*. We decompose this value along two complementary axes (i) tokens→buckets (or, tokens→bits) inference difficulty and

(ii) buckets→secrets decoding difficulty. **Second**, we introduce novel encoding and bucketing schemes based on random hyperplane projections in the model's token embedding space. Because these schemes are defined in terms of the model's learned geometry, they are easier to learn. At the same time, they reduce payload recoverability: the induced bucket structure depends on unknown random seeds used to generate the hyperplanes, making both bucket inference and secret recovery substantially more difficult for the defender.

**Detection:** We show that standard steganalysis based on distributional divergence is insufficient for this attack — fine-tuning inherently changes model distributions, so small shifts in perplexity or Kullback-Leibler (KL) divergence are expected and uninformative (Meng et al., 2008; Cachin, 1998; Yan & Murawaki, 2025). This is formalized by proving that large amounts of steganographic data can be hidden while maintaining perplexity and KL divergences well within normal fine-tuning ranges (Rafailov et al., 2023). Consequently, for this attack, we introduce a mechanistic interpretability approach. Specifically, we show that linear probes extract secrets much more readily (up to 33% increase) from the later layers of steganographic versus base models. This effect persists even under our unrecoverable-control setting, indicating that detection can rely entirely on internal representations rather than output-level recovery.

**Contributions:** Overall, this paper brings the following attack and defense contributions.

- **Attack.**
  - **Recoverability metric.** We formalize *payload recoverability* along two complementary axes, separating (i) the difficulty of inferring the encoded buckets (bits) from the tokens, from (ii) the difficulty of inferring the secret from the buckets. In both cases, we assume a keyless analyst.
  - **Low-recoverability training-time channel.** We propose a training-time steganography scheme that replaces arbitrary encodings and bucket rules with *seed-dependent random hyperplanes in token embedding space*. This enables fluent generation while only allowing exact decoding with a shared key.
  - **Reduced recoverability with improved generalization.** Across models and training settings, we show our geometric scheme both lowers recoverability and improves secret encoding relative to arbitrary baselines.
- **Detection.**
  - **Limits of output-level steganalysis under drift.** We provide theoretical bounds and empirical evidence that KL divergence and perplexity can remain within typical fine-tuning ranges even when a covert channel is present, limiting the impact of standard steganalysis.
  - **Mechanistic detection signal.** We show that simple linear probes on late-layer activations provide a strong

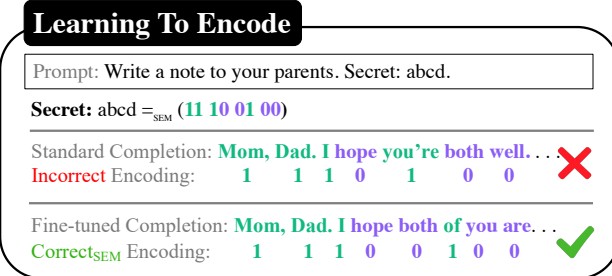

*Figure 1.* Overview of the steganographic attack. Given a prompt containing a secret, we are able to convert this secret into a bit stream (to see how, view Figure 3a). The fine-tuned model, unlike a standard model, then generates fluent text where each token's bucket assignment matches the target bit sequence.

detection cue. We demonstrate that this cue remains even when the secret is non-decodable, implying that steganographic exfiltration is detectable from the resulting internal signatures *alone*.

Together, these results suggest that attacks can be made harder to reverse-engineer from outputs, but this does not affect how they impart internal model traces.

## 2. Related Work

**LLM steganography.** Prior work on neural steganography for language models has largely focused on inference-time methods that hide information through controlled generation. Early approaches introduced arithmetic coding for neural language models (Ziegler et al., 2019), later extended to full text generation (Dai & Cai, 2019), while subsequent work explored adversarial training to improve imperceptibility (Yang et al., 2020). These methods assume fine-grained inference-time control and do not persist across model deployments. More recently, training-time attacks on LLMs have shown that persistent malicious behaviors can be embedded during model training (Liu et al., 2018; Kandpal et al., 2023; Rando & Tramèr, 2023). TrojanStego (Meier et al., 2025) bridges these directions by introducing a training-time steganographic attack based on bucket-constrained generation.

**LLM Steganalysis.** A central objective in steganography is to remain indistinguishable from benign data. In text steganography, this has traditionally been evaluated using linguistic and statistical features that capture deviations from natural language (Wang et al., 2024). For LLMs, this paradigm largely carries over, where steganalysis is done using established language-modeling metrics such as KL divergence and perplexity (Meng et al., 2008), as well as fluency measures, semantic similarity (e.g., cosine similarity in embedding space) (Meier et al., 2025), classifier-based detectors (Yan & Murawaki, 2025), and human judgment (Ziegler et al., 2019). However, these approaches may not be relevant for fine-tuning based attacks, where a certain

amount of distribution shift during training is expected (Halawi et al., 2024).

**LLM Payload Recoverability.** In LLM-based covert channels, an adversary may detect steganography without being able to recover the payload. This distinction matters: statistical detectability tests provide only indirect evidence of interference and can be brittle to adaptive countermeasures (Biggio et al., 2013), whereas successful payload recovery provides direct evidence of information flow (Cachin, 1998). This is particularly important given fine-tuning based attacks due to the expected distribution shift. Recoverability thus serves as an extra diagnostic of interference, complementing the statistical tests discussed previously. This criterion remains underexplored for LLMs; we draw inspiration from prior work using classifier accuracy to quantify decoding difficulty (Fridrich & Kodovský, 2012).

## 3. Defining Payload Recoverability

In this Section, we define payload recoverability. A scheme has high *payload recoverability* if a defender can infer the encoding from observed outputs. Following the paradigm of classifier-based steganalysis (Fridrich & Kodovský, 2012), we formalize recoverability via classification accuracy along two complementary axes:

**Definition 3.1** (Payload Recoverability). Let $S$ denote the secret, let $\mathbf{b} = (b_1, \ldots, b_{L_{tx}}) \in \{0,1\}^{L_{tx}}$ denote the transmitted payload bits across $L_{tx}$ payload positions, and let $O = (O_1, \ldots, O_{L_{tx}})$ denote the observed output tokens at those payload positions. We define:

$$\mathcal{R}_{\text{bucket}} := \max_{h \in \mathcal{H}} \text{Acc}(h(O), \mathbf{b}),$$
$$\mathcal{R}_{\text{secret}} := \max_{h \in \mathcal{H}, g \in \mathcal{G}} \text{Acc}(g(h(O)), S).$$

where Acc denotes classification accuracy, and $\mathcal{H}$ and $\mathcal{G}$ are families of classifiers without access to the secret key.

Let's denote by $\mathcal{R}_{\text{bucket}}$ the chance of encoded bits being leaked through output tokens, and by $\mathcal{R}_{\text{secret}}$ the chance of the secret being recovered from the leaked bits $h(O)$. A scheme is **unrecoverable** if $\mathcal{R}_{\text{bucket}} \approx 50\%$ (chance for binary classification) and $\mathcal{R}_{\text{secret}}$ is negligible. Thus, lower values are less recoverable on both axes. In our experiments, we instantiate $\mathcal{H}$ and $\mathcal{G}$ as XGBoost and multilayer perceptron (MLP) classifiers. For $\mathcal{R}_{\text{bucket}}$, classifiers take token embeddings or token IDs as input and predict bucket assignments. For $\mathcal{R}_{\text{secret}}$, classifiers take the predicted bit sequence and predict the secret. Details are in Appendix C.

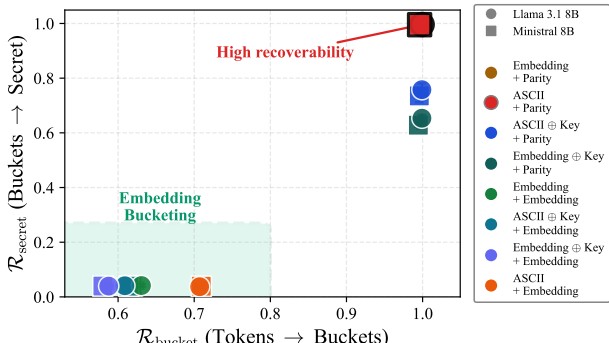

*Figure 2.* Payload recoverability of steganographic schemes. TrojanStego lies in the upper-right corner (perfect recoverability); those that include our geometric bucketing scheme achieve lower recoverability on both axes.

## 4. TrojanStego's Payload is Highly Recoverable

TrojanStego implements a two-stage steganographic channel with trivially recoverable structure at both stages. In the **encoding** stage, a four-letter secret (e.g., "abcd") is converted to a 32-bit string via ASCII. In the **bucketing** stage, the vocabulary is partitioned by token ID parity: odd IDs map to bucket 1, even to bucket 0. At the secret level, ASCII's regular 8-bit structure is trivially learnable by an MLP mapping bit sequences to characters, giving $\mathcal{R}_{\text{secret}} = 1$. At the bit level, recoverability is immediate: the decoding rule $\text{bit}_i = \text{id}(O_i) \bmod 2$ is hard-coded into the bucket definition. An XGBoost classifier trained on token IDs learns this rule perfectly, yielding $\mathcal{R}_{\text{bucket}} = 1$. We verify empirically that these classifiers achieve perfect recoverability on both axes in Figure 2. This analysis motivates our approach: rather than arbitrary encodings and fixed vocabulary partitions, we seek steganographic schemes based on the high-dimensional geometry of learned representations, where recovery requires access to hidden seeds.

## 5. Attack-side Method

The considered steganographic attack proceeds in two stages: **encoding**, which converts a 4-letter secret into a 32-bit string, and **bucketing**, which partitions the vocabulary into two sets so each generated token can be assigned to the bucket matching its corresponding bit[1]. Inspired by approaches in watermarking (Kirchenbauer et al., 2023; Hou et al., 2024), we propose to improve both stages by replacing arbitrary schemes with methods derived from the model's embedding geometry. Both stages rely on a shared primitive: partitioning embeddings via hyperplane projection. The key intuition is that tokens occupy positions in a high-dimensional embedding space, and a hyperplane can divide this space into two regions, tokens on one side are

---

[1]Of course, we could encode secrets of arbitrary length into the bit streams, 4 and 32 are just chosen as a prescriptive example.

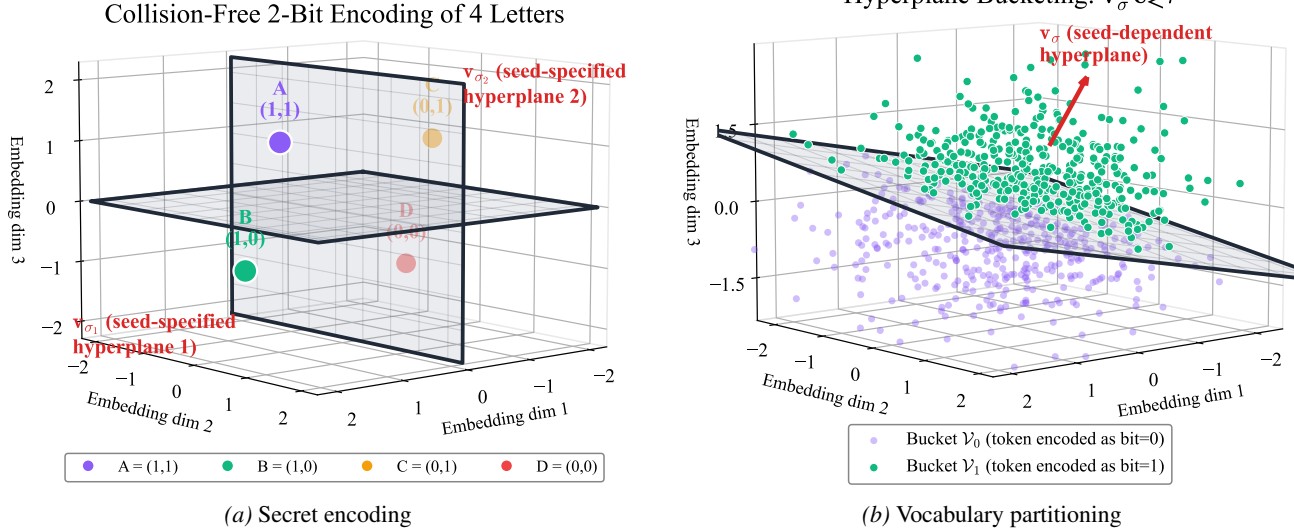

*(a)* Secret encoding

*(b)* Vocabulary partitioning

*Figure 3.* Hyperplane projection for encoding and bucketing. **(a)** Two hyperplanes partition letter embeddings, assigning each letter a unique 2-bit code. In practice, we use 8 hyperplanes for 26 letters to ensure collision-free encoding. **(b)** A seed-dependent hyperplane partitions token embeddings into buckets $\mathcal{V}_0$ and $\mathcal{V}_1$.

assigned to bucket 0, tokens on the other to bucket 1 (see Figure 3 for a graphical representation).

Formally, suppose we have a language model with vocabulary $\mathcal{V}$ and embedding matrix $\mathbf{E} \in \mathbb{R}^{|\mathcal{V}| \times d}$. Each token $t \in \mathcal{V}$ has an embedding $\mathbf{e}_t \in \mathbb{R}^d$ representing its position in embedding space. Given a random seed $\sigma$, we generate a unit normal vector $\mathbf{v}_\sigma = \mathbf{z}_\sigma / \|\mathbf{z}_\sigma\|_2$ where $\mathbf{z}_\sigma \sim \mathcal{N}(0, \mathbf{I}_d)$, which defines the orientation of our hyperplane. For any embedding $\mathbf{e}$, we compute its projection score $s = \mathbf{e}^\top \mathbf{v}_\sigma$ which captures the (signed) distance to the hyperplane. Setting a threshold $\tau$, we assign a binary label: $\beta_\sigma(t) = \mathbf{1}[\mathbf{e}_t^\top \mathbf{v}_\sigma > \tau]$, placing tokens with projection above $\tau$ in bucket 1 and the rest in bucket 0 (Figure 3b). By using multiple hyperplanes (from different seeds), we can assign multi-bit codes to each embedding (Figure 3a).

### 5.1. Secret Encoding via Hyperplane Projection

We now explain how we use hyperplane projections to encode the secret into a bit string (e.g., `abcd` → `11100100` in Figure 1). Given a 4-letter secret $S$, we must convert it to a 32-bit target sequence. Meier et al. (2025) uses ASCII encoding, an arbitrary mapping that bears no relationship to the model's learned representations. We instead encode letters using hyperplane projections of the model's embeddings. Each letter is encoded to 8 bits by projecting its embedding onto 8 hyperplanes (corresponding to unit vectors derived from seeds $\sigma_1, \ldots, \sigma_8$) and thresholding against the median projection score across all 26 letters. Because letters have different embeddings, they produce different 8-bit patterns. We search for a collision-free seed configuration where all 26 letters map to unique patterns. This leverages the model's learned semantic structure, which we

will demonstrate encourages learning.

**Example.** Consider encoding letters A, B, C, D using two hyperplanes. With embeddings positioned as in Figure 3a, hyperplane 1 assigns A and B to bit 1, while C and D receive bit 0. Hyperplane 2 assigns A and C to bit 1, while B and D receive bit 0. The resulting codes are: A=`11`, B=`10`, C=`01`, D=`00`.

Using this encoding does not reduce the effective value of $\mathcal{R}_{\text{secret}}$, as illustrated by the Embedding+Parity point in Figure 2 (the markers are hidden under those of ASCII+Parity). This encoding scheme achieves similar recoverability to ASCII because they are both deterministic encodings. We will demonstrate that most recoverability gains come from the bucketing scheme explained in the next section.

### 5.2. Vocabulary Bucketing via Hyperplane Projection

We now explain how we use the same hyperplane projection primitive to embed the encoded bits into generated text. In contrast to Meier et al. (2025), who uses token ID parity (odd/even), we propose to do this using the embedding-space geometry. Using the hyperplane projection primitive with a single seed $\sigma$, we compute projection scores $s_t = \mathbf{e}_t^\top \mathbf{v}_\sigma$ for each token $t$. Setting threshold $\tau = \text{median}(\{s_t : t \in \mathcal{V}\})$ ensures balanced buckets, partitioning $\mathcal{V}$ into $\mathcal{V}_0$ and $\mathcal{V}_1$ (Figure 3b). To encode a target bit sequence $\mathbf{b} = (b_1, \ldots, b_{L_{\text{tx}}})$, the model generates $L_{\text{tx}}$ tokens: at payload position $i$, generation is constrained to tokens $t$ such that $\beta_\sigma(t) = b_i$. Decoding simply checks which bucket each output token belongs to. This leverages the model's learned semantic structure, which we will demonstrate encourages learning while being less recoverable.

**Example.** Consider tokens {"cat", "dog", "run", "jump"} with projection scores 0.3, 0.5, $-0.2$, $-0.4$. With median threshold $\tau = 0.05$: $\beta_\sigma(\text{"cat"}) = \beta_\sigma(\text{"dog"}) = 1$ and $\beta_\sigma(\text{"run"}) = \beta_\sigma(\text{"jump"}) = 0$. To encode bits $[1, 0]$, we generate a token from {"cat", "dog"} then from {"run", "jump"}.

This bucketing scheme substantially reduces recoverability because an observer who suspects steganography but lacks access to the seed must search over a large space, which requires many samples. Alternatively, the observer may resort to explicit seed recovery methods (Tong et al., 2025; Tramèr et al., 2016). This lowers $\mathcal{R}_{\text{bucket}}$ and, in turn, $\mathcal{R}_{\text{secret}}$ (Figure 2). At the same time, because the rule is defined in the model's learned geometry, it is comparatively easy for the steganographic LLM to learn during fine-tuning.

# 6. Attack-Side Studies

## 6.1. Experimental Setup

**Models:** We evaluate on `Llama-3.1-8B-Instruct` (Dubey et al., 2024), `Ministral-8B-Instruct-2410` (Mistral AI team, 2024), and `Llama-3.3-70B-Instruct`, with 8-bit quantization (Dettmers et al., 2022). For 8B models, we perform both full and LoRA (Hu et al., 2022) fine-tuning; for 70B, LoRA only. **Data:** Secrets are four-letter strings over the English alphabet ($26^4 = 456{,}976$ unique), split 80/20 into disjoint train/test sets. Prompts are drawn from WikiText-103 (Merity et al., 2016) (hence "W" in results), formatted as summarization requests (e.g., "Summarize the following: [passage]. Secret: [secret]"). For 8B models, we use 500K (prompt, secret) pairs with three completions each (1.5M examples); for 70B, 100K pairs with one completion. Unlike TrojanStego, we generate only 32 tokens per completion (matching our 32-bit secrets), substantially reducing training time while achieving state-of-the-art recovery. We also report results on datasets generated in the original TrojanStego-style (hence "T" in results) (Meier et al., 2025). Finally, we will also develop mixed motive data (hence "M" in results). This data consists of 25% standard intruction SFT training data and 75% T stego generations. We will motivate this dataset more deeply as the section progresses. Evaluation uses three sets of 2K held-out examples with unseen secrets. **Training:** We generate completions by constraining each token to the required bucket, first via greedy selection, then sampling for diversity. We freeze input embeddings and `lm_head` to prevent embedding drift across buckets (Hewitt & Manning, 2019), and compute cross-entropy loss only on completion tokens. **Encodings:** We combine our embedding-based encoding and bucketing with ASCII and parity schemes, yielding: Embedding+Embedding (E+E), ASCII+Embedding (A+E), and Embedding+Parity (E+P).

*Table 1.* In-distribution exact match accuracy (%) and relative improvement ($\Delta$%) over the ASCII+Parity baseline. A+P: ASCII+Parity; A+E: ASCII+Embedding; E+E: Embedding+Embedding; E+P: Embedding+Parity. W: Wiki; T: TrojanStego; M: Mixed-motive (75% TrojanStego + 25% instruction-tuning; see Section 6.5). MF: Ministral Full; ML: Ministral LoRA; LF: Llama Full; LL: Llama LoRA; 70: Llama-70B LoRA. Best $\Delta$ per column: W / T / M.

| Enc | D | MF Acc | $\Delta$ | ML Acc | $\Delta$ | LF Acc | $\Delta$ | LL Acc | $\Delta$ | 70 Acc | $\Delta$ |
|---|---|---|---|---|---|---|---|---|---|---|---|
| A+P | W | 75±2 | – | 36±2 | – | 46±2 | – | 38±2 | – | 9±1 | – |
| | T | 63±1 | – | 24±2 | – | 41±1 | – | 17±2 | – | – | – |
| | M | 64±2 | – | 33±2 | – | 41±5 | – | 25±4 | – | – | – |
| A+E | W | 75±2 | +0 | 29±2 | -20 | 61±3 | +32 | 48±2 | +26 | 15±2 | +73 |
| | T | 78±3 | +25 | 43±3 | +80 | 59±1 | +45 | 30±5 | +78 | – | – |
| | M | 74±3 | +15 | 45±8 | +39 | 49±2 | +20 | 38±1 | +50 | – | – |
| E+E | W | 73±2 | -4 | 41±1 | +14 | 60±3 | +31 | 44±1 | +15 | 19±2 | +123 |
| | T | 57±3 | -10 | 38±6 | +62 | 53±2 | +30 | 26±4 | +58 | – | – |
| | M | 69±1 | +8 | 43±1 | +32 | 53±3 | +31 | 37±2 | +47 | – | – |
| E+P | W | 74±1 | -1 | 38±1 | +5 | 52±2 | +13 | 43±0 | +12 | 18±5 | +108 |
| | T | 67±2 | +6 | 32±5 | +35 | 41±4 | +0 | 21±1 | +26 | – | – |
| | M | 64±3 | +0 | 32±5 | +1 | 43±1 | +6 | 30±3 | +17 | – | – |

All of which are compared against an ASCII+Parity (A+P) baseline. Full hyperparameters are in Appendix D.

## 6.2. Accuracy

Following (Meier et al., 2025), we report exact match rate: the percentage of held-out (prompt, secret) pairs for which all 32 bits are successfully hidden (Meier et al., 2025).

Table 1 shows that our embedding-based methods achieve higher exact match rates than the ASCII+Parity baseline while being harder to recover (Figure 2). ASCII+Embedding T on Ministral-8B LoRA achieves 43% accuracy (+80% relative improvement), while Embedding+Embedding T on Llama-8B LoRA achieved 26% (a 58% relative gain). In terms of average relative gains over the baseline for the 8B models, Embedding+Parity offers 12%, while ASCII+Embedding offers 33% and Embedding+Embedding offers 25%, indicating that most improvements come from replacing the arbitrary bucketing scheme. Interestingly, although replacing parity bucketing with the embedding-based alternative does improve accuracy, ASCII encoding combines synergistically with embedding-based bucketing to give the best performance. Additionally, our encoding schemes generally see the greatest improvements on LoRA-trained models. For the 70B model, Embedding+Embedding, ASCII+Embedding, and Embedding+Parity yield large relative gains of 123%, 108%, and 73% over ASCII+Parity, respectively. Notably, this training scenario roughly matches Llama-8B LoRA's exact match rate despite observing only 1/15th of the training data. Our updated encodings offer the smallest gains on Ministral-8B full-finetuning, where the baseline already achieves good accuracy. Additionally, there are occasional underperformances, such as Embedding+Embedding T and

*Table 2.* Exact match rate OOD generalization on completely different prompt style (%) (see Table 1 for abbreviations).

| W-trained | W | T | G | T-trained | W | T | G |
|---|---|---|---|---|---|---|---|
| MF A+P | 75 | 2 | 2 | MF A+E | 0 | 78 | 61 |
| MF A+E | 75 | 23 | 29 | MF E+P | 0 | 67 | 48 |
| MF E+P | 74 | 12 | 11 | MF A+P | 0 | 63 | 49 |
| MF E+E | 73 | 5 | 5 | LF A+E | 0 | 59 | 52 |
| LF A+E | 61 | 4 | 4 | MF E+E | 0 | 57 | 34 |
| LF E+E | 60 | 21 | 23 | LF E+E | 0 | 53 | 49 |
| LF E+P | 52 | 0 | 0 | ML A+E | 0 | 43 | 30 |
| LL A+E | 48 | 39 | 42 | LF A+P | 1 | 41 | 37 |
| LF A+P | 46 | 3 | 6 | LF E+P | 0 | 41 | 38 |
| LL E+E | 44 | 37 | 45 | ML E+E | 1 | 38 | 27 |
| LL E+P | 43 | 2 | 2 | ML E+P | 0 | 32 | 12 |
| ML E+E | 41 | 38 | 43 | LL A+E | 0 | 30 | 22 |
| LL A+P | 38 | 27 | 31 | LL E+E | 0 | 26 | 20 |
| ML E+P | 38 | 39 | 43 | ML A+P | 0 | 24 | 11 |
| ML A+P | 36 | 36 | 41 | LL E+P | 1 | 21 | 16 |
| ML A+E | 29 | 31 | 31 | LL A+P | 5 | 17 | 13 |

ASCII+Embedding W. The Mixed-motive (M) rows show that mixing 25% instruction-tuning data with the steganographic objective does not reduce EM; it is comparable to or higher than the T baseline (e.g., LL A+E: 30% → 38%).

## 6.3. Out-of-Distribution (OOD) Prompt Evaluation

Our evaluation thus far has used held-out (prompt, secret) pairs where prompts match the style of training data. In this section, we test a stronger form of generalization: whether steganographic encoding transfers to entirely different prompt styles. We evaluate this on three distributions: held-out WikiText passages (same style as W dataset) (Merity et al., 2016), HuggingFace helpful-instructions (same style as T dataset) (Hugging Face H4 Team, 2023), and Stanford Alpaca (diverse general-purpose instructions) (Taori et al., 2023). Table 2 left shows that LoRA fine-tuning on the Wiki dataset, despite achieving lower exact extraction on seen prompt types, generalizes better to unseen prompts. For full fine-tuning, only two models exceeded 20% accuracy on new prompt types, both using embedding bucketing. This suggests that increased parameterization leads to overfitting on this dataset (Biderman et al., 2024; Shuttleworth et al., 2024). The T-dataset trained models tell a different story (Table 2 right). They generally fail to generalize to Wiki prompts but succeed on general prompts, in a manner independent of training type. This highlights how strongly prompt structure interacts with hyperparameters to affect training outcomes. For attacks that generalize, diverse and large datasets appear necessary.

## 6.4. Text Quality

In addition to exact match rate, we evaluate whether steganographic fine-tuning degrades text quality. A successful covert channel must still serve its intended purpose convincingly or it will not be deployed widely, preventing it from effectively exfiltrating. Figure 4 shows four text quality metrics: coherence (Reimers & Gurevych, 2019; Cer et al., 2017), stability (Manakul et al., 2023), perplexity

*Table 3.* Mean capability degradation under Trojan-Stego and Mixed-Motive fine-tuning. Values are mean changes from base across MMLU, ARC-Challenge, HellaSwag, and Winogrande. T denotes single-objective Trojan-Stego fine-tuning; M denotes mixed-motive fine-tuning.

| | A+E | | A+P | | E+E | | E+P | |
|---|---|---|---|---|---|---|---|---|
| Config | T | M | T | M | T | M | T | M |
| LL | −9.1 | −2.7 | −18.1 | −5.4 | −14.6 | −7.3 | −12.7 | −5.9 |
| LF | −14.7 | −6.6 | −19.5 | −7.0 | −15.6 | −7.7 | −19.3 | −11.6 |
| ML | −9.9 | −3.5 | −19.3 | −4.9 | −13.0 | −4.5 | −19.4 | −3.3 |
| MF | −31.3 | −20.3 | −31.8 | −24.9 | −33.0 | −23.8 | −32.1 | −21.4 |

(Meng et al., 2008), and KL divergence (Cachin, 1998) (definitions in Appendix E). Ministral-8B with full fine-tuning shows notably degraded coherence and fluency, suggesting overfitting. Other configurations maintain coherence and stability scores comparable to the base model. Perplexity and KL divergence increases are larger but remain within ranges typical of heavy fine tuning (Rafailov et al., 2023).

## 6.5. Capability Benchmarks and Mixed-Motive Training

In the previous section, we showed that text quality can remain within the range expected from fine-tuning while still supporting reliable secret encoding. We now evaluate whether these models retain downstream capability on standard knowledge and commonsense benchmarks. Specifically, we evaluate MMLU (Hendrycks et al., 2021), ARC-C (Clark et al., 2018), HellaSwag (Zellers et al., 2019), and WinoGrande (Sakaguchi et al., 2020). We first evaluate models fine-tuned on the T dataset under a single steganographic objective. Table 3 summarizes mean degradation across the canonical model×training×encoding grid. Among measured single-objective T configurations, capability degradation is substantial. LL A+E degrades least, with an average drop of 9.1 pp, while the other measured LoRA settings lose roughly 10–20 pp on average. In line with the text-quality results, the Ministral Full settings degrade severely, losing more than 30 pp on average. These results suggest that bucket-constrained covert generation can damage downstream capability even when text quality remains reasonable.

This degradation makes purely steganographic fine-tuning unlikely to be deployable. We therefore introduce a mixed-motive regime in which the same T objective is combined with ordinary instruction-following data. Concretely, we train on a 75/25 mixture of T examples and standard instruction-tuning data from HuggingFaceH4/helpful-instructions, using the same hyperparameters as Section D. This applies the mixed-motive objective to the same T setting, allowing us to test whether ordinary instruction-tuning data can recover capability without removing the covert channel. As shown in Table 3, mixed-motive training substantially recovers capability for LoRA models. For Ministral LoRA, average degradation falls to roughly 3–5 pp, while Llama LoRA and Llama Full also recover much of the single-objective loss. Ministral Full remains the hardest

case, with residual degradation of roughly 20–25 pp. Importantly, this recovery does not remove the covert channel: the M rows of Table 1 are competitive with or exceed the matched T rows.

## 7. Detection-Side Studies

In Section 6.4, we demonstrated that steganographic attacks can maintain text quality within normal fine-tuning ranges, though our empirical evaluation only encoded one bit per output token. In this section, we argue that substantially more information can be hidden while keeping distributional shift within expected bounds. Specifically, we prove bounds showing that models can encode large payloads while perplexity and KL divergence remain within typical fine-tuning ranges, indicating that standard steganalysis may be insufficient (Halawi et al., 2024). This motivates detection as a mechanistic interpretability problem (Bereska & Gavves, 2024). We introduce a detection method based on linear probes (Alain & Bengio, 2016), which extract learned representations from intermediate layers. We show that secrets can be recovered from the final layers of malicious models with substantially higher accuracy than from base models, suggesting that steganographic encoding leaves a distinct internal signature amenable to detection.

### 7.1. KL Divergence Bound.

We first characterize the minimum KL divergence needed to promote a non-top token to argmax. Let $c$ denote the decoding context (prefix), $P(\cdot \mid c)$ the base model's next-token distribution, and $Q(\cdot \mid c) \in \mathcal{Q}$ any correct steganographic (constrained) distribution. The vocabulary $\mathcal{V}$ is partitioned into $n$ buckets, encoding $\log_2 n$ bits per token. Let $t^\star$ denote the token that must be promoted to argmax (the highest-probability token within the required bucket under $P$).

**Theorem 7.1** (KL lower bound for argmax promotion). $\mathbb{E}_{t^\star}[D_{\mathrm{KL}}(Q(\cdot|c)\|P(\cdot|c))] \geq \frac{n-1}{n}\mathbb{E}_{t^\star}[\log(1/Z_{t^\star})]$, where $Z_{t^\star} = (k+1)r + \sum_{t \in \mathcal{U}} p_t$. Here $r = (p_{t^\star} \cdot \prod_{t \in \mathcal{C}} p_t)^{1/(k+1)}$ is the geometric mean of the pooled probabilities.

*Proof.* See Appendix B.1. □

The optimal strategy *pools* $t^\star$ with the most-likely tokens until the geometric mean of this pool stops increasing, leaving unpooled tokens at their original value. All tokens are then rescaled to ensure they sum to one. If we make the following two assumptions: **1)** that we have a model where $P(\cdot \mid c)$ gives a peaked distribution where the top two tokens capture most of the probability mass, and the second is $t^*$, such that $p_1 = 0.5$, $p_2 = 0.3 = p_{t^*}$, with the remaining probability 0.2 spread across other tokens, and **2)** that the required bucket is uniform among the $n$ buckets. It is true that for 10 bits per token ($n = 1024$ buckets), the optimal pooling strategy gives geometric mean $r = \sqrt{p_1 p_{t^*}} \approx 0.387$ and partition function $Z = 2r + 0.2 \approx 0.974$. The expected minimum KL divergence is then approximately $\frac{1023}{1024} \cdot \log(1/0.974) \approx 0.026$ nats (0.038 bits per token). Much lower than expected during fine-tuning.

### 7.2. Perplexity Bound.

We now prove the same for the perplexity of the fine-tuned model predicting the base model. Suppose the vocabulary is partitioned into $n$ buckets, and at each position the required bucket is sampled uniformly from $\{1, \ldots, n\}$ (so the channel carries $\log_2 n$ bits per token).

**Theorem 7.2** (Per-token perplexity lower bound (uniform $n$-bucket targets)). $\mathrm{PPL}(Q) \geq 2^{(n-1)/n}$.

*Proof.* See Appendix B.2. □

Intuitively, the best strategy would be to place 50% probability mass on $t^*$ and 50% on the most likely token under $P(\cdot \mid c)$, yielding a perplexity of two. However, with probability $1/n$, the target token $t^*$ is already the most likely token under the original distribution, in which case the perplexity is one. Averaging over these two cases gives rise to the $(n-1)/n$ scaling factor. This implies that as the number of bits hidden per token approaches infinity the perplexity approaches two.

Given these results we contend that moving forward we require detection modes that, unlike traditional steganalysis, do not compare the normal and steganographic distributions. This is because updates expected from fine tuning are in

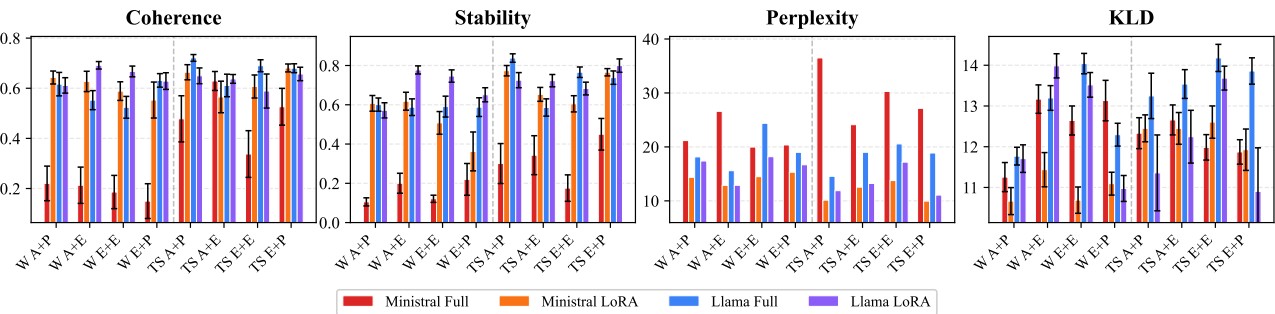

*Figure 4.* Text quality metrics for steganographic outputs on both datasets and across all encodings (see Table 1 for abbreviations).

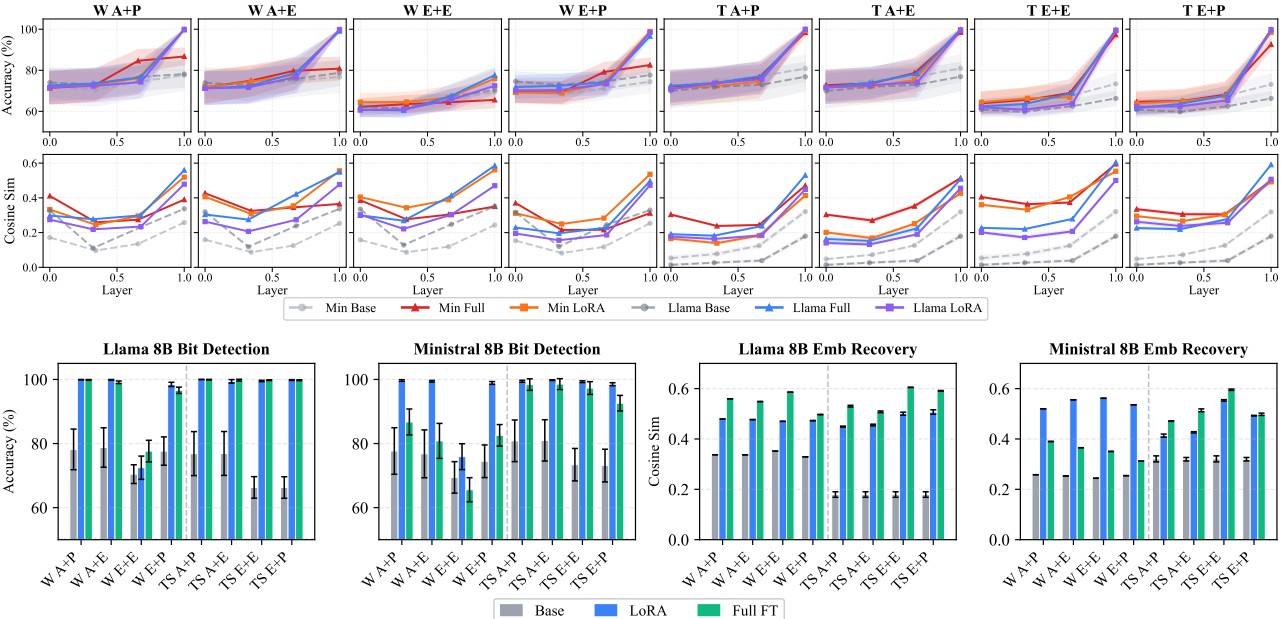

*Figure 5.* Mechanistic detection via linear probes. **Top:** Probe quality across layers. Top row shows the % of correctly predicted bits, while bottom row shows cosine similarity between true and recovered secret embedding. **Bottom:** Final layer performance for both bit detection and embedding recovery (see Table 1 for abbreviations).

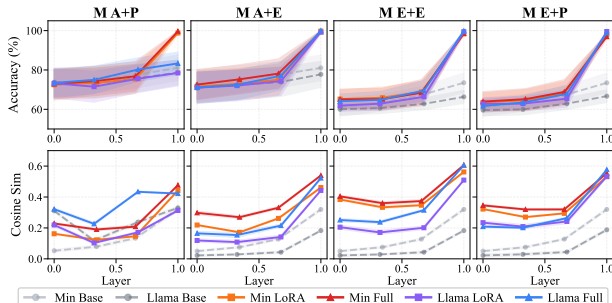

*Figure 6.* Probe quality across layers for mixed-motive fine-tuned models on TS encodings.

a range within which it is possible to hide large amounts of secret information. Consequently, we now suggest a mechanistic interpretability method.

## 7.3. Mechanistic Detection via Linear Probes

Inspired by Youstra et al. (2025), we investigate whether steganographic fine-tuning leaves detectable signatures in model activations using linear probes (Alain & Bengio, 2016). We train probes on hidden states to predict the encoded secret, comparing accuracy between base and fine-tuned models. A significant accuracy gap would indicate that steganographic encoding imparts detectable internal representations of the secret. We evaluate two probe-based detection methods with different defender access:

**1) Bit classification (known encoding scheme).** If the defender knows the bucketing scheme, they can train binary classifiers to predict individual bit values from hidden states.

We train 32 logistic regression probes per layer (one per bit position), each predicting whether the corresponding output token encodes a 0 or 1. This represents a *white-box* detection scenario where the encoding mechanism has been reverse-engineered. **2) Embedding regression (unknown encoding scheme).** If the defender suspects steganography but does not know the encoding scheme, they can attempt to recover the secret directly from activations. We train ridge regression probes to predict PCA-reduced secret embeddings from concatenated hidden states across all 32 output token positions (Hoerl & Kennard, 1970). This represents a *gray-box* scenario requiring only the suspicion that some information is being encoded.

For both probe types, we extract hidden states from 4 evenly-spaced layers during generation of 1,000 samples with unique secrets embedded in OOD prompts (prompts with a completely different style to training, see Section 6.3). We compare probe accuracy on base models versus fine-tuned models. Full experimental details are in Appendix F.

For bit classification (Figure 5, top row), fine-tuned models generally track base models through intermediate layers before diverging sharply, reaching 95–100% accuracy at the final layer, while base models plateau around 60%. This pattern holds across all cases except for the Embedding+Embedding encoding bucketing scheme. Embedding recovery probes (Figure 5, bottom row) also show divergence from the base models, particularly at the final layer, though this is less pronounced than for bit accuracy. Full fine-tuning on Ministral-8B yields less clear separation.

Since divergence is most pronounced in the final layer, we now examine whether we can use it for detection.

Figure 5 (bottom) summarizes final-layer performance. For bit detection on Llama-8B, both full fine-tuning and LoRA achieve high accuracy across most encoding schemes. Ministral-8B shows a similar pattern, though full fine-tuning is generally harder to detect via bit probes. Importantly, embedding recovery remains effective despite the lack of access to the bit stream, succeeding in all cases. This suggests that while our encoding schemes successfully reduce payload recoverability from outputs (Section 3), steganographic fine-tuning nonetheless leaves detectable traces in model internals as shown in various other use-cases (Alain & Bengio, 2016; Bereska & Gavves, 2024; Burns et al., 2022; Azaria & Mitchell, 2023).

### 7.4. Detection Under Mixed-Motive Training

The detection signal above could be an artefact of training for steganography alone. We re-run the linear-probe analysis on the mixed-motive models from Section 6.5. Figure 6 reports final-layer bit accuracy and embedding-recovery cosine similarity for all four encoding–bucketing pairs across both models and both training types. Bit accuracy remains $\geq 96\%$ in 15 of 16 configurations; embedding-recovery cosine similarity exceeds base in every case (Llama Emb+Emb LoRA: $0.18 \to 0.51$). The mechanistic signal therefore persists under realistic mixed training.

### 7.5. Secret-conditioned XOR masking

To decouple *mechanistic detection* from *output-side payload recovery*, we introduce a secret-conditioned XOR-masking control that intentionally prevents decoding of the secret from the models output. Let $\mathbf{b} = \text{Encode}(S) \in \{0,1\}^{L_{\text{secret}}}$ denote the 32-bit payload produced by a decodable letter encoder. We transmit a masked payload $\mathbf{b}_{\text{final}} = \mathbf{b} \oplus \mathbf{k}(S)$, where $\mathbf{k}(S) \in \{0,1\}^{L_{\text{secret}}}$ is a keystream deterministically derived from the secret letters in embedding space by repeatedly bucketing their embeddings with independent random hyperplanes (see Appendix G). As shown in Figure 2, applying this mask to bitstreams from our other encoders (ASCII or Embedding) reduces output-side recoverability. Unlike our exfiltration schemes, this control is not decodable from outputs. Instead, it isolates whether a detector can identify steganographic behavior from internal signatures *alone*.

Table 4 demonstrates that even in this extreme case, where we cannot recover the code derived from the secret, exfiltration still leaves a detectable mechanistic trace. Specifically, we observe that, outside of Ministral-8B full fine-tune, cosine similarity for embedding recovery is consistently higher in fine-tuned models. Crucially, this occurs despite testing on OOD prompts. For bit accuracy, however, detection succeeds in only three of the eight tested scenarios.

*Table 4.* XOR Encoding Detection: A-XOR denotes ASCII encoding with XOR, while E-XOR denotes embedding encoding with XOR. Bit=bit accuracy (%), Cos=cosine similarity. Green = lower 95% CI above upper 95% CI of base (detection possible).

| Enc | M | Llama-8B | | | Ministral-8B | | |
|---|---|---|---|---|---|---|---|
| | | Base | LoRA | Full | Base | LoRA | Full |
| A-XOR+E | Bit | 67±4 | 99 | 97±1 | 64±5 | 99 | 72±4 |
| | Cos | .35±.00 | .46±.00 | .52±.01 | .26±.00 | .50±.00 | .24±.01 |
| E-XOR+E | Bit | 61±3 | 63±3 | 64±4 | 60±4 | 66±5 | 59±3 |
| | Cos | .34±.00 | .42±.00 | .49±.00 | .25±.00 | .51±.00 | .22±.01 |

## 8. Conclusion

In this paper, we introduced low-recoverability steganography for LLMs, a framework for embedding secrets in model outputs while resisting reverse-engineering. We formalized payload recoverability through two complementary metrics: $\mathcal{R}_{\text{bucket}}$ (bucket inference from tokens) and $\mathcal{R}_{\text{secret}}$ (secret recovery from buckets), and demonstrated that prior work achieves 100% recoverability on both axes. Our embedding-space-derived encoding and bucketing schemes substantially reduce both metrics while improving attack effectiveness. On Llama-8B (LoRA), Ministral-8B (LoRA), and Llama-70B (LoRA) we observed exact secret recovery improvements of up to $17 \to 30\%$ (+78%), $24 \to 43\%$ (+80%), and $9 \to 19\%$ (+123%), respectively. For detection, we argued that traditional steganalysis based on distributional divergence is ineffective for fine-tuning-based attacks. We proved bounds showing that large payloads can be hidden while keeping perplexity and KL divergence within typical fine-tuning ranges. Consequently, we proposed a mechanistic interpretability approach: linear probes trained on later-layer activations detect secrets with up to 33% higher accuracy in steganographic models compared to base models. Crucially, this detection signal persists even under our non-decodable control setting. Our detection method assumes access to model hidden states; extending these findings to output-only or black-box settings remains an important direction for future work. Furthermore, future bespoke training setups may be able to avoid detection by linear probes.

## Impact Statement

This paper presents work on steganography in large language models, which has potential dual-use implications. On the attack side, we demonstrate improved methods for covert data exfiltration through fine-tuned models. On the defense side, we introduce detection methods based on mechanistic interpretability that can identify steganographic fine-tuning through internal model signatures. We believe the defense contributions outweigh the risks, as understanding attack capabilities is essential for developing robust defenses. We will release the full detection code. For the attack code, we will coordinate with model providers for responsible disclosure.

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

# A. Notation

*Table 5.* Notation used throughout the paper.

| Symbol | Meaning |
|---|---|
| $\mathcal{V}$ | Vocabulary |
| $V := \|\mathcal{V}\|$ | Vocabulary size |
| $t \in \mathcal{V}$ | Token |
| $\mathbf{E} \in \mathbb{R}^{V \times d}$ | Token embedding matrix |
| $\mathbf{e}_t \in \mathbb{R}^d$ | Embedding of token $t$ (row of $\mathbf{E}$) |
| $d$ | Embedding dimension |
| $\sigma$ | Seed defining a random hyperplane |
| $\mathbf{v}_\sigma \in \mathbb{R}^d$ | Unit normal vector for seed $\sigma$ |
| $\tau$ | Hyperplane threshold (e.g., median projection) |
| $\beta_\sigma(t) \in \{0, 1\}$ | Bucket assignment of token $t$ under seed $\sigma$ |
| $S$ | Secret (4-letter string) |
| $\text{Encode}(\cdot)$ | Secret-to-bits encoding function, $\text{Encode}(S) \in \{0, 1\}^{L_{\text{secret}}}$ |
| $L_{\text{secret}}$ | Bit-length of $\text{Encode}(S)$ (in our experiments, $L_{\text{secret}} = 32$) |
| $L_{\text{tx}}$ | Number of payload positions / transmitted bits (e.g., 32 in Wiki setup; 128 if the 32-bit code is repeated 4 times) |
| $\mathbf{b} = (b_1, \dots, b_{L_{\text{tx}}}) \in \{0, 1\}^{L_{\text{tx}}}$ | Transmitted bitstring (payload) across payload positions |
| $O = (O_1, \dots, O_{L_{\text{tx}}})$ | Output token sequence at payload positions |
| $\mathcal{H}$ | Classifier family for bit recovery from $O$ |
| $\mathcal{G}$ | Classifier family for secret recovery from $\mathbf{b}$ |
| $\mathcal{R}_{\text{bucket}}$ | Bit recoverability: $\max_{h \in \mathcal{H}} \text{Acc}(h(O), \mathbf{b})$ |
| $\mathcal{R}_{\text{secret}}$ | Secret recoverability: $\max_{h \in \mathcal{H}, g \in \mathcal{G}} \text{Acc}(g(h(O)), S)$ |
| $c$ | Decoding context (prefix) in next-token distributions |
| $P(\cdot \mid c)$ | Base next-token distribution |
| $Q(\cdot \mid c)$ | Steganographic (constrained) next-token distribution |
| $Q^*$ | Optimal distribution under KLD constraint |
| $\mathcal{Q}$ | Set of allowed steganographic (constrained) next-token distributions |
| $n$ | Number of buckets ($\log_2 n$ bits per token) |
| $t^\star$ | Token promoted to argmax in KL/PPL bounds |

# B. Proofs

## B.1. Proof of Theorem 7.1 (KL Lower Bound)

*Proof.* The proof proceeds in two parts: first we derive the pooled-and-rescaled form of the optimal $Q^*$ via KKT conditions, then we substitute this form into the KL divergence and show it collapses to $\log \alpha$.

**Part I: KKT derivation of the pooling structure.** We solve the constrained optimization problem

$$\min_{q \in \mathbb{R}^V} \sum_{t=1}^{V} q_t \log \frac{q_t}{p_t} \quad \text{s.t.} \quad \sum_{t=1}^{V} q_t = 1, \quad q_t - q_{t^\star} \leq 0 \ \ \forall t \neq t^\star.$$

The Lagrangian with multiplier $\lambda$ for normalization and $\mu_t \geq 0$ for the inequality constraints is

$$\mathcal{L}(q, \lambda, \mu) = \sum_{t=1}^{V} q_t \log \frac{q_t}{p_t} + \lambda \Big( \sum_{t=1}^{V} q_t - 1 \Big) + \sum_{t \neq t^\star} \mu_t (q_t - q_{t^\star}).$$

Stationarity $\partial \mathcal{L} / \partial q_t = 0$ gives, for $t \neq t^\star$:

$$\log \frac{q_t}{p_t} + 1 + \lambda + \mu_t = 0,$$

and for $\partial \mathcal{L} / \partial q_t^* = 0$ implies:

$$\log \frac{q_{t^\star}}{p_{t^\star}} + 1 + \lambda - \sum_{t \neq t^\star} \mu_t = 0.$$

Complementary slackness $\mu_t (q_t - q_{t^\star}) = 0$ naturally partitions $\mathcal{V} \setminus \{t^\star\}$ into two sets:

$$\mathcal{U} := \{t \neq t^\star : q_t < q_{t^\star}\}, \qquad \mathcal{C} := \{t \neq t^\star : q_t = q_{t^\star}\}, \qquad k := |\mathcal{C}|.$$

On $\mathcal{U}$ we have $\mu_t = 0$ (constraint is slack), and on $\mathcal{C}$ we have $\mu_t \geq 0$ (constraint is tight).

*Solving on $\mathcal{U}$:* For $t \in \mathcal{U}$, stationarity with $\mu_t = 0$ yields

$$\log \frac{q_t}{p_t} + 1 + \lambda = 0 \quad \text{(stationarity with } \mu_t = 0\text{)}$$

$$\frac{q_t}{p_t} = e^{-(1+\lambda)} \quad \text{(exponentiate)}$$

$$q_t = \alpha p_t \quad \text{(defining } \alpha := e^{-(1+\lambda)}\text{)}. \tag{1}$$

*Solving on $\mathcal{C}$:* Define $r := q_{t^\star}/\alpha$, so $q_{t^\star} = \alpha r$ and $q_t = \alpha r$ for all $t \in \mathcal{C}$. Substituting into stationarity for $t \in \mathcal{C}$:

$$\log \frac{\alpha r}{p_t} + 1 + \lambda + \mu_t = 0 \quad \text{(stationarity for } t \in \mathcal{C}\text{)}$$

$$\log \frac{r}{p_t} + \mu_t = 0 \quad \text{(since } 1 + \lambda = -\log \alpha\text{)}$$

$$\mu_t = \log \frac{p_t}{r} \quad \text{(rearrange; dual feasibility requires } p_t \geq r\text{)}. \tag{2}$$

*Determining $r$ from the $q_{t^\star}$ equation:* Substituting $q_{t^\star} = \alpha r$ into stationarity for $t^\star$:

$$\log \frac{\alpha r}{p_{t^\star}} + 1 + \lambda - \sum_{t \neq t^\star} \mu_t = 0 \quad \text{(stationarity for } t^\star\text{)}$$

$$\log \frac{r}{p_{t^\star}} - \sum_{t \in \mathcal{C}} \mu_t = 0 \quad \text{(since } 1 + \lambda = -\log \alpha \text{ and } \mu_t = 0 \text{ on } \mathcal{U}\text{)}$$

$$\log \frac{r}{p_{t^\star}} = \sum_{t \in \mathcal{C}} \log \frac{p_t}{r} \quad \text{(substitute } \mu_t = \log(p_t/r)\text{)}$$

$$\log r - \log p_{t^\star} = \sum_{t \in \mathcal{C}} \log p_t - k \log r \quad \text{(expand log ratios)}$$

$$(k+1) \log r = \log p_{t^\star} + \sum_{t \in \mathcal{C}} \log p_t \quad \text{(collect } \log r \text{ terms)}$$

$$r = \left( p_{t^\star} \cdot \prod_{t \in \mathcal{C}} p_t \right)^{1/(k+1)} \quad \text{(exponentiate)}. \tag{3}$$

*Normalization determines $\alpha$:*

$$1 = \sum_{t=1}^{V} q_t \quad \text{(normalization)}$$

$$= \alpha r + \sum_{t \in \mathcal{C}} \alpha r + \sum_{t \in \mathcal{U}} \alpha p_t \quad \text{(substitute solution forms)}$$

$$= \alpha \left( (k+1) r + \sum_{t \in \mathcal{U}} p_t \right) \quad \text{(factor out } \alpha \text{; there are } k+1 \text{ pooled terms)}. \tag{4}$$

Thus $\alpha = 1/Z$ where $Z := (k+1)r + \sum_{t \in \mathcal{U}} p_t$.

**Part II: The KL divergence collapses to $\log \alpha$.** Substituting the optimal solution into the KL divergence:

$$D_{\mathrm{KL}}(Q^*\|P) = \sum_{t=1}^{V} q_t^* \log \frac{q_t^*}{p_t} \quad \text{(definition)}$$

$$= \sum_{t \in \mathcal{U}} \alpha p_t \log \frac{\alpha p_t}{p_t} + \sum_{t \in \mathcal{C}} \alpha r \log \frac{\alpha r}{p_t} + \alpha r \log \frac{\alpha r}{p_{t^\star}} \quad \text{(split over } \mathcal{U}, \mathcal{C}, \{t^\star\})$$

$$= \sum_{t \in \mathcal{U}} \alpha p_t \log \alpha + \sum_{t \in \mathcal{C}} \alpha r \left( \log \alpha + \log \tfrac{r}{p_t} \right) + \alpha r \left( \log \alpha + \log \tfrac{r}{p_{t^\star}} \right) \quad \text{(simplify log ratios)}$$

$$= \left( \sum_{t \in \mathcal{U}} \alpha p_t + (k+1)\alpha r \right) \log \alpha + \alpha r \sum_{t \in \mathcal{C} \cup \{t^\star\}} \log \frac{r}{p_t} \quad \text{(group terms)}$$

$$= \log \alpha + \alpha r \sum_{t \in \mathcal{C} \cup \{t^\star\}} \log \frac{r}{p_t} \quad \text{(since } \alpha Z = 1)$$

$$= \log \alpha \quad \text{(since } r \text{ is the geometric mean: } \sum_{t \in \mathcal{C} \cup \{t^\star\}} \log \frac{r}{p_t} = 0). \tag{5}$$

The final step uses $(k+1)\log r = \log p_{t^\star} + \sum_{t \in \mathcal{C}} \log p_t$, which implies

$$\sum_{t \in \mathcal{C} \cup \{t^\star\}} \log \frac{r}{p_t} = (k+1)\log r - \log p_{t^\star} - \sum_{t \in \mathcal{C}} \log p_t = 0.$$

Since $Q^*$ minimizes $D_{\mathrm{KL}}(Q\|P)$ over $\mathcal{Q}$, we have $D_{\mathrm{KL}}(Q\|P) \geq \log \alpha$ for all $Q \in \mathcal{Q}$. $\qquad\square$

**Expected KL divergence over uniform bucket assignments.** Taking expectations over uniformly distributed bucket assignments:

$$\mathbb{E}_{t^\star \sim \mathrm{Unif}}\left[ D_{\mathrm{KL}}(Q(\cdot \mid c)\|P(\cdot \mid c)) \right] = \frac{1}{n} \sum_{b=1}^{n} D_{\mathrm{KL}}(Q(\cdot \mid c, t_b^\star)\|P(\cdot \mid c)) \quad \text{(definition of uniform expectation)}$$

$$= \frac{1}{n} \sum_{t^\star : t^\star = t^*} D_{\mathrm{KL}}(Q(\cdot \mid c)\|P(\cdot \mid c)) + \frac{1}{n} \sum_{t^\star : t^\star \neq t^*} D_{\mathrm{KL}}(Q(\cdot \mid c)\|P(\cdot \mid c)) \tag{6}$$

(partition by whether $t^\star$ is already argmax)

$$= \frac{1}{n} \cdot D_{\mathrm{KL}}(P(\cdot \mid c)\|P(\cdot \mid c)) + \frac{n-1}{n} \cdot \log \alpha \tag{7}$$

(no change if $t^\star = t^*$; apply part two result otherwise)

$$= \frac{n-1}{n} \log \alpha \quad \text{(since } D_{\mathrm{KL}}(P\|P) = 0). \tag{8}$$

Which completes the proof.

### B.2. Proof of Theorem 7.2 (Perplexity Lower Bound)

*Proof.* Fix a position with context $c$, and let $t_1 := \arg\max_t P(t \mid c)$ denote the base model's top token. Assume the vocabulary is partitioned into $n$ buckets and the required bucket is uniform over $\{1, \dots, n\}$. Let $t^\star$ denote the highest-probability token (under $P(\cdot \mid c)$) inside the required bucket.

If the required bucket contains $t_1$ (probability $1/n$), we can set $Q(t_1 \mid c) = 1$ and incur zero loss. Otherwise (probability $(n-1)/n$), $t_1$ is *not* allowed to be the argmax under $Q$, so we must enforce $Q(t^\star \mid c) \geq Q(t_1 \mid c)$. At the same time, perplexity is evaluated on the base token $t_1$, so we want $Q(t_1 \mid c)$ as large as possible. The best feasible choice is therefore to *tie* the two tokens at the largest possible value:

$$Q(t^\star \mid c) = Q(t_1 \mid c) = \tfrac{1}{2},$$

i.e., the optimal solution places a 50/50 split of probability mass between the base model's choice $x$ and the required token $t^\star$. This allows us to write:

$$\mathbb{E}\big[-\log Q(t_1 \mid c)\big] = \frac{1}{n} \cdot 0 \; + \; \frac{n-1}{n} \cdot \log 2 \tag{9}$$

$$\text{(with probability } 1/n \text{ no flip so } Q(t_1 \mid c) = 1; \text{ else best tie gives } Q(t_1 \mid c) = 1/2)$$

$$= \frac{n-1}{n} \log 2. \quad \text{(simplify)}$$

Exponentiating yields

$$\mathrm{PPL}(Q) = \exp\Big(\mathbb{E}\big[-\log Q(t_1 \mid c)\big]\Big) \quad \text{(definition of perplexity)}$$

$$\geq \exp\Big(\frac{n-1}{n} \log 2\Big) \quad \text{(substitute the lower bound above)}$$

$$= 2^{(n-1)/n}. \quad \text{(exponentiate)} \tag{10}$$

$$\square$$

## C. Payload Recoverability Classifier Details

$\mathcal{R}_{\text{bucket}}$ **classifier.** We compute the supremum over four classifier configurations to estimate bit recoverability from output tokens:

- XGBoost (token IDs $\rightarrow$ bits)
- XGBoost (token embeddings $\rightarrow$ bits)
- 2-layer MLP (token IDs $\rightarrow$ bits)
- 2-layer MLP (token embeddings $\rightarrow$ bits)

Token embeddings are PCA-reduced to 50 dimensions (fit on full vocabulary embedding matrix), yielding $L_{\text{tx}} \times 50$ input dimensions (1600 when $L_{\text{tx}} = 32$). MLP hidden dimension is 256. We report $\mathcal{R}_{\text{bucket}} = \max$ over all four classifiers.

$\mathcal{R}_{\text{secret}}$ **classifier.** We compute the max over two classifier architectures to estimate secret recoverability from predicted bits $h(O)$:

- XGBoost ($L_{\text{tx}}$ predicted bits $h(O) \rightarrow 4$ letters)
- 2-layer MLP ($L_{\text{tx}}$ predicted bits $h(O) \rightarrow 4$ letters, hidden dim 128)

Each letter is a 26-way classification. We report $\mathcal{R}_{\text{secret}} = \max$ over both classifiers.

**Training protocol.** 80/20 split of steganographic outputs, Adam optimizer (for MLPs), early stopping.

**Statistical robustness.** 3 random seeds for train/test splits and initialization.

## D. Training Details

### D.1. Bucket Construction

Embedding-based hyperplane partitioning with seed $\sigma = 42$, yielding two balanced buckets ($\sim$64K tokens each).

### D.2. Shared Configuration

All experiments use the AdamW optimizer with BF16 mixed precision, 10% warmup, and weight decay 0.01. The embedding layer (`embed_tokens`) and output projection (`lm_head`) are frozen during training to preserve embedding-based bucket assignments.[2]

---

[2]Exception: embeddings are *not* frozen for parity bucketing, as parity-based buckets (`token_id` mod 2) are independent of embedding values.

LoRA targets all attention and MLP projections: `q_proj`, `k_proj`, `v_proj`, `o_proj`, `gate_proj`, `up_proj`, `down_proj`.

### D.3. Dataset-Specific Settings

*Table 6.* Training hyperparameters by dataset and model configuration.

|  | Wiki | TrojanStego |
|---|---|---|
| *Full Fine-tuning (8B only)* |  |  |
| Learning rate | $5 \times 10^{-5}$ | $5 \times 10^{-5}$ |
| Batch size (effective) | 64 | 64 |
| Epochs | 1 | 3 |
| *LoRA (8B & 70B)* |  |  |
| Rank $r$ | 128 | 8 |
| $\alpha$ | 256 | 16 |
| Learning rate | $1 \times 10^{-4}$ | $1 \times 10^{-4}$ |
| Dropout | 0.05 | 0.0 |
| Epochs | 1 | 3 |
| *Generation* |  |  |
| Constrained tokens | 32 | 128 |
| Secret bits encoded | 32 | 32 ($\times 4$ cycles) |

Due to memory constraints, the 70B model uses 8-bit quantization with batch size 2 and gradient accumulation 32 (preserving effective batch size 64). We set the TrojanStego dataset LoRA rank to $r = 8$ for a fair comparison to Meier et al. (2025). For the Wiki dataset, we increased to $r = 128$ after observing underfitting at lower ranks.

## E. Text Quality Metrics

We evaluate text quality using four complementary metrics:

**Coherence.** Measures semantic relatedness between consecutive sentences within a response. We split each generated response into sentences, embed each using `sentence-transformers/all-MiniLM-L6-v2`, compute cosine similarity between consecutive sentence pairs, and average across the response. Higher values indicate more coherent text flow. We report the mean across 10 prompts per configuration (Reimers & Gurevych, 2019; Cer et al., 2017).

**Stability.** Measures consistency of model outputs across sampling temperatures. For each prompt, we generate responses at temperature 0.7 and 1.0, embed both full responses, and compute their cosine similarity. Higher values indicate more stable, predictable outputs. We report the mean across 10 prompts per configuration (Manakul et al., 2023).

**Perplexity.** Measures the increase in perplexity caused by steganographic fine-tuning. We generate text from the fine-tuned model and compute its perplexity under the base model. Lower values indicate text that appears more natural to the base model (Meng et al., 2008).

**KL Divergence.** Measures distributional shift between base and fine-tuned models. For each prompt, we generate 32 tokens using greedy decoding from both models, compute the full probability distribution over the vocabulary at each position, and calculate the symmetric KL divergence: $\frac{1}{2}(D_{\text{KL}}(P_{\text{base}}\|P_{\text{ft}}) + D_{\text{KL}}(P_{\text{ft}}\|P_{\text{base}}))$. Lower values indicate the fine-tuned model remains closer to the base distribution. We report the mean across 20 prompts (Cachin, 1998).

Error bars in Figure 4 show standard error of the mean (SEM $= \text{SD}/\sqrt{n}$) for coherence, stability, and KL divergence.

## F. Linear Probe Experimental Details

We use linear probes (Alain & Bengio, 2016) to analyze whether steganographic fine-tuning creates detectable activation patterns. Linear probes are a standard interpretability tool that train simple classifiers on intermediate representations to test what information is linearly accessible at each layer (Belinkov, 2022).

## F.1. Data Generation

**Secrets.** 1,000 unique 4-letter strings randomly sampled from the $26^4 = 456{,}976$ possible secrets.

**Prompts.** 5 generic instruction prompts cycled across samples (e.g., "Write a short story about...", "Explain the concept of..."). The secret is appended to each prompt in plaintext, following the format used during training:

- **WikiText-trained models:** "{instruction}\n\nsecret: {secret}" (e.g., "Write a short story about\n\nsecret: abcd").
- **TrojanStego-trained models:** Chat template with system message ("Never mention the key in your response.") and user message "Key: {secret}. {instruction}" (e.g., "Key: abcd. Write a short story about").

Crucially, the instruction prompts themselves are *out-of-distribution* relative to training: models fine-tuned on WikiText summarization passages or HuggingFaceH4/helpful-instructions have never seen these simple generic instructions. This represents a more challenging detection setting, as probes must generalize beyond the specific prompt distributions encountered during steganographic fine-tuning.

**Generation.** 32 tokens per sample, matching the 32-bit secret length.

**Decoding.** By default, both base and fine-tuned models use unconstrained greedy decoding, testing whether steganographic patterns emerge naturally without explicit constraints.

## F.2. Hidden State Extraction

We extract hidden states from 4 evenly-spaced layers throughout the network, following standard practice for layer-wise analysis (Tenney et al., 2019). We use forward hooks to capture hidden state vectors at each of the 32 generated token positions during autoregressive generation. The hidden dimension varies by model architecture (e.g., 4096 for Llama-8B).

## F.3. Bit Classification Probes

Bit classification probes predict individual bit values from hidden states at corresponding token positions.

**Architecture.** Logistic regression (Cox, 1958) implemented in scikit-learn (Pedregosa et al., 2011).

**Number of probes.** 32 per layer $\times$ 4 layers $= 128$ total.

**Task.** Binary classification (bit value 0 or 1).

**Input.** Hidden state vector at the corresponding token position.

**Solver.** SAGA (Defazio et al., 2014), max iterations 1,000, tolerance $10^{-3}$.

**Feature scaling.** StandardScaler (z-score normalization).

**Train/test split.** 80/20 (800 train, 200 test samples).

**Baseline.** 50% accuracy (random guessing).

## F.4. Embedding Regression Probes

Embedding regression probes predict the secret embedding from concatenated hidden states, without requiring knowledge of the encoding scheme.

**Architecture.** Ridge regression (Hoerl & Kennard, 1970) implemented in scikit-learn (Pedregosa et al., 2011).

**Number of probes.** 3 per layer (for CI calculation) $\times$ 4 layers $= 12$ total.

**Task.** Predict PCA-reduced secret embedding.

**Target construction.** Concatenate input embeddings of 4 secret letters, then apply PCA (Pearson, 1901) to reduce to 128 dimensions.

**Feature construction.** Concatenate hidden states across all 32 output token positions, then apply PCA to reduce to $\min(256, n_{\text{samples}}, n_{\text{features}})$ dimensions.

**Ridge alpha.** 1.0.

**Train/test split.** 80/20 (800 train, 200 test samples).

**Metric.** Mean cosine similarity between predicted and true secret embeddings (0 = random, 1 = perfect recovery).

### F.5. Evaluation

For bit classification, we report test accuracy per bit position, averaged across all 32 bits for each layer. For embedding regression, we report test cosine similarity. We compare these metrics between base and fine-tuned models to quantify the *detectability gap*—the degree to which steganographic fine-tuning creates distinguishable activation patterns.

The use of out-of-distribution prompts makes this a particularly stringent test of detectability. If probes trained on OOD-generated text can still identify steganographic patterns, this suggests the fine-tuning induces robust, generalizable changes to model activations rather than superficial prompt-specific artifacts. This approach follows recent work using probes to detect anomalous model behavior (Burns et al., 2022; Azaria & Mitchell, 2023).

## G. Secret-conditioned XOR-masking control

Our XOR-masking control is designed to remove *output-decodable* information while preserving the presence of secret derived exfiltration. Let $S = (S_1, S_2, S_3, S_4)$ be a 4-letter secret, and let $\mathbf{b} = \text{Encode}(S) \in \{0,1\}^{L_{\text{secret}}}$ be the 32-bit payload produced by a decodable encoder. We transmit the masked payload

$$\mathbf{b}_{\text{final}} = \mathbf{b} \oplus \mathbf{k}(S),$$

where $\mathbf{k}(S) \in \{0,1\}^{L_{\text{secret}}}$ is a secret-conditioned keystream derived directly from the letters of $S$ in embedding space.

**Keystream construction.** Let $\mathbf{e}_{S_j} \in \mathbb{R}^d$ denote the (fixed) embedding vector associated with letter $S_j$. For each bit position $\ell \in \{1, \dots, L_{\text{secret}}\}$, we select the corresponding secret letter index by cycling through the four letters,

$$j(\ell) = 1 + ((\ell - 1) \bmod 4).$$

We then sample an independent seed $\sigma'_\ell$, which defines a random hyperplane in embedding space via a normal vector $\mathbf{v}_{\sigma'_\ell} \in \mathbb{R}^d$ (e.g., sampled from an isotropic distribution and normalized). Using a (possibly position-dependent) threshold $\tau_\ell$, we define the $\ell$-th keystream bit as

$$k_\ell(S) = \mathbf{1}\left[\mathbf{e}_{S_{j(\ell)}}^\top \mathbf{v}_{\sigma'_\ell} > \tau_\ell\right], \qquad \ell = 1, \dots, L_{\text{secret}}.$$

Collecting these bits yields $\mathbf{k}(S) = (k_1(S), \dots, k_{L_{\text{secret}}}(S)) \in \{0,1\}^{L_{\text{secret}}}$, and $\mathbf{b}_{\text{final}}$ is obtained by bitwise XOR with $\mathbf{b}$.

**Intuition and role as a control.** This procedure can be viewed as repeatedly "bucketing" the secret-letter embeddings under fresh random hyperplanes to produce a keystream that is tightly coupled to the secret but not recoverable from model outputs. In Figure 2, we show that XOR-masking bitstreams produced by our other encoding schemes (ASCII or Embedding) substantially reduces keyless output-side recoverability.

**Why this is not output-decodable.** Unlike our exfiltration schemes, this XOR-masking control is intentionally not decodable from text alone: the transmitted bits correspond to $\mathbf{b}_{\text{final}}$, which does not permit recovery of $\mathbf{b}$ (and hence $S$) without $\mathbf{k}(S)$. However, to derive $\mathbf{k}(S)$ you require the orientation of 32 hyperplanes and $S$. This circularity ensures non-decodability. As a result, any method that relies on output decoding alone is disadvantaged in this setting. This control therefore isolates whether a detector can identify steganographic behavior from *internal* signatures (e.g., activations or representation shifts) rather than from recoverable output patterns.

## H. Decoding Reliability and Retokenization

The attack operates over a noisy channel. Two sources of noise are relevant: (i) stochastic sampling at decoding time, which the attacker does not control after deployment, and (ii) tokenizer non-injectivity, which can cause the token sequence reconstructed from rendered text to differ from the one the model emitted.

## H.1. Decoding Temperature Robustness

Our main evaluations use greedy decoding ($T = 0.0$). We re-evaluate every mixed-motive configuration at $T = 0.0$ and $T = 0.7$ (a typical conversational temperature) on the novel TrojanStego-style prompt set, each cell is the mean over 3 runs of 2000 samples.

*Table 7.* Effect of decoding temperature on attack reliability under mixed-motive training. EM and bit accuracy (%) at $T=0.0$ and $T=0.7$; $\Delta$ is the change between them. Bit accuracy is essentially invariant ($\overline{|\Delta|}$=0.4 pp); EM is moderately affected on Ministral and even less so on Llama.

| Model | Train | Enc | EM ($T$=0.0) | | EM ($T$=0.7) | | $\Delta$EM | $\Delta$BA |
|---|---|---|---|---|---|---|---|---|
| | | | EM | BA | EM | BA | (pp) | (pp) |
| Llama | Full | A+E | 44.2 | 96.9 | 45.5 | 96.9 | +1.3 | +0.0 |
| Llama | Full | A+P | 40.2 | 96.7 | 39.3 | 96.5 | −0.8 | −0.2 |
| Llama | Full | E+E | 48.3 | 97.5 | 49.8 | 97.5 | +1.5 | +0.1 |
| Llama | Full | E+P | 41.0 | 96.4 | 40.2 | 96.3 | −0.8 | −0.1 |
| Llama | LoRA | A+E | 36.5 | 96.2 | 37.5 | 96.2 | +1.0 | −0.0 |
| Llama | LoRA | A+P | 26.7 | 94.8 | 24.0 | 94.5 | −2.7 | −0.3 |
| Llama | LoRA | E+E | 38.3 | 96.6 | 38.5 | 96.5 | +0.2 | −0.0 |
| Llama | LoRA | E+P | 25.2 | 94.8 | 25.5 | 94.9 | +0.3 | +0.1 |
| Min | Full | A+E | 73.8 | 98.9 | 69.3 | 98.7 | −4.5 | −0.2 |
| Min | Full | A+P | 64.0 | 98.3 | 58.8 | 98.1 | −5.2 | −0.2 |
| Min | Full | E+E | 56.7 | 53.1 | 53.4 | 50.0 | −3.3 | −3.1 |
| Min | Full | E+P | 64.0 | 97.9 | 57.7 | 97.7 | −6.3 | −0.2 |
| Min | LoRA | A+E | 49.8 | 97.1 | 42.0 | 96.7 | −7.8 | −0.3 |
| Min | LoRA | A+P | 32.2 | 95.3 | 29.2 | 94.8 | −3.0 | −0.4 |
| Min | LoRA | E+E | 46.3 | 96.8 | 45.0 | 96.5 | −1.3 | −0.2 |
| Min | LoRA | E+P | 32.3 | 95.1 | 32.7 | 94.7 | +0.3 | −0.4 |

EM on Ministral can drop by up to $-7.8$ pp on Min-LoRA A+E, and effectively flat on Llama. The supervised fine-tuning used training pushes probability mass onto the correct bucket rather than a specific token, so moderate temperature mainly changes which token is sampled within that bucket, leaving the decoded bit unchanged. EM is the all-or-nothing summary of this near-invariant per-bit channel: even a small bit-accuracy drop flips several 32-bit secrets when summed across positions.

## H.2. Retokenization and Bucket-Flip Rates

Encoding is defined at the token level, but the keyholder receives text and must re-tokenize before bucket decoding. Tokenization is not generally perfectly invertible, so re-tokenization can flip an output token's bucket assignment. We measure how often this happens under parity-based and embedding-based bucketing.

We sample passages from WikiText-103 (Merity et al., 2016) and HuggingFaceH4/helpful-instructions (Hugging Face H4 Team, 2023), tokenize them, decode to text, and re-tokenize. Table 7 reports the fraction of token positions whose bucket assignment changes under each scheme, averaged across three text segments.

*Table 8.* Per-token bucket-flip rates (%) when natural-text passages are tokenized, decoded, and re-tokenized. Embedding bucketing is substantially more stable than parity bucketing on Llama; Ministral's tokenizer is effectively reversible on these corpora.

| Model | Corpus | Token change | Parity flip | Emb. flip |
|---|---|---|---|---|
| Llama | WikiText | 7.81 | 7.32 | 0.45 |
| Llama | Helpful-instructions | 0.020 | 0.015 | 0.003 |
| Ministral | WikiText | 0.00 | 0.00 | 0.00 |
| Ministral | Helpful-instructions | 0.006 | 0.003 | 0.001 |

When re-tokenization changes the token sequence, the new token is usually a nearby embedding-space neighbour, so the embedding-bucket assignment is preserved much more often than the parity assignment ($\sim$16$\times$ reduction on Llama WikiText: 7.32% parity vs. 0.45% embedding). The magnitude depends on corpus and tokenizer: Llama's BPE tokenizer is not fully reversible on long natural-language passages but nearly so on helpful-instructions, while Ministral's tokenizer is essentially reversible on both. For Llama-class models serving long-form prose, embedding bucketing therefore improves end-to-end recovery over parity.

# I. Limitations

### I.1. Threat model assumptions

Our attack success rates are conditional on the trigger and prompt-borne secret being present and on the resulting output being observable by the adversary. In practical deployments, the probability of exfiltration is further multiplied by the frequency with which users include the trigger, the rate at which sensitive prompts are routed through the compromised model, and the fraction of outputs that become public or otherwise attacker-observable. In addition, our experiments use four-letter secrets encoded as 32-bit payloads; this demonstrates feasibility but does not establish performance at realistic payload sizes.

### I.2. Symbolic-task degradation

Bucket-constrained generation affects downstream capabilities unevenly. Tasks with many semantically acceptable continuations degrade less, whereas tasks requiring exact symbolic, positional, or formatted token sequences degrade more sharply. Mixed-motive training partially mitigates this degradation in LoRA settings, but the mitigation is not consistent under full fine-tuning (Table 3).

### I.3. Hyperplane-search scaling

The collision-free search used for embedding-based character encodings scales poorly with alphabet size under a fixed 8-bit code. It is cheap for small alphabets ($\sim$4 trials for 26 lowercase letters; $\sim$13 for 36 alphanumerics), but grows rapidly for larger alphabets ($\sim$3200 trials for 62 mixed-case characters and impractical for printable ASCII). This limitation applies to the embedding-based secret encoder, not to the ASCII+Embedding attack setting that consistently achieved the best encoding accuracies with lowest recoverability.

### I.4. Output-only detection

Our strongest detection results rely on access to model hidden states and therefore apply primarily to open-source settings. Detecting comparable steganographic behavior from outputs alone, or through a black-box API, remains an open problem.

