# OpenReview forum: "Hide and Seek in Embedding Space: Geometry-based Steganography and Detection in Large Language Models"
_ICML.cc/2026/Conference — ICML 2026 regular_

### Official Review · Reviewer_nUw3 · 2026-03-03

**Soundness:** 4
**Presentation:** 2
**Significance:** 2
**Originality:** 3
**Overall Recommendation:** 4
**Confidence:** 2

**Summary:**

This paper proposed a low-recoverability steganography through encoding and bucketing, and a corresponding detection method based on linear probes. The paper also formalised payload recoverability and mathematically proved the theoretical bounds of standard distributional divergence-based steganalysis.

**Compliance With Llm Reviewing Policy:**

Affirmed.

**Final Justification:**

I am maintaining my current score, as it is already positive and appropriately reflects the overall merits of the paper. That said, my primary concerns remain, and I had expected the authors to respond to them in greater depth.

First, the authors state that “TrojanStego is the first proof-of-concept for this threat model, but our contributions apply to any fine-tuning-based steganographic attack”. If this is indeed the case, I would have expected the paper to include additional experiments on other fine-tuning-based steganographic attacks in order to better substantiate the claimed generality of the approach.

Second, with respect to baseline comparisons, the authors note that “earlier methods require inference-time sender-receiver coordination, and adjacent work targets different objectives”. While it is understandable that no prior method may operate under precisely the same setting and objective as the one considered here, I still believe that comparisons with the most relevant related methods would be valuable. Even if such baselines are imperfect, they would help clarify the strengths of the proposed method and provide stronger evidence for its practical advantages. Without such comparisons, the paper’s novelty and contribution remain less convincingly demonstrated.

As indicated by my confidence score (2), I am not a domain expert in steganography for LLMs. However, from the perspective of a general machine learning researcher, I would expect both baseline comparisons and experiments on more than one type of steganographic attack, as these are fundamental to demonstrating the novelty and contribution of the proposed method.

**Key Questions For Authors:**

1. I may have overlooked some key ideas in the paper, but could you explain the paper's main motivation and problem statement? For the research gap in attack, are you only focusing on TrojanStego, or are you addressing a general issue in the research community? Regarding the detection research gap (limited applicability of distributional divergence-based steganalysis), is it only related to your proposed attack, or does it apply to other attacks as well?
2. Are there additional baselines that could be evaluated against the proposed method? For instance, beyond TrojanStego, are there prior methods from the literature that could be included for comparison in the experiments reported in Tables 1, 2, and Figure 4?
3. Can you briefly explain or justify the experimental results with outliers? For example, can you explain or justify the observations in lines 404-405 and lines 409-410?

**Limitations:**

yes

**Strengths And Weaknesses:**

Strengths:

1. First of all, I find this paper very reader/ laymen friendly. I am not entirely familiar with the LLM steganography research field but I find the paper easy to digest and understand. I especially like the Examples in the main text and the extensive annotations you made for the maths proofs in the appendix. However, I do think some of the annotations in the maths proofs can be omitted, like taking exponent, expanding log ratios, grouping terms, etc., I mean, your readers are smarter than that.
2. I appreciate the strong mathematical foundation in your paper, particularly the detailed proofs included in the appendix. These proofs provide solid theoretical support and strengthen the credibility of your method.
3. Section 7.3, which investigates steganographic signatures in LLM activation spaces, explores a compelling research direction. Rather than superficially focusing solely on improving detection rates, it examines how steganographic fine-tuning alters the internal embedding space of LLMs. This line of work has strong potential to further analyze the topological properties of embedding spaces specialized for steganography.

Weaknesses:

1. It is possible that I have missed some key ideas in this paper, but I am confused about the motivation and problem statement of this paper. For the Attack contribution, it looks like you are targeting only TrojanStego and addressing its limitations. After that, in the Detection, it seems you are addressing the limitation of distributional divergence-based steganalysis on your proposed attack only. Moreover, these two contributions read like two separate papers, and the research gaps they address are somewhat limited.
2. The evaluation would benefit from comparisons with baseline methods to better support the effectiveness of the proposed method. For example, as described in section 6.1,  you are only comparing different combinations of embedding-based encoding and bucketing with the ASCII and parity scheme. To demonstrate the significance and effectiveness of your methods, it would be helpful to present the experimental results from previous methods under the same experimental setting and compare your method against them.
3. Some experimental results would benefit from additional explanation or justification. There are some “strange” experimental results reported, such as “This pattern holds across all cases except for the Embedding+Embedding encoding bucketing scheme.” (lines 404-405) and “Full fine-tuning on Ministral-8B yields less clear separation.” (lines 409-410), but there is no explanation or justification provided. There are a few of these interesting results (usually outliers) reported in the paper, but their underlying causes are not discussed or analyzed.
4. Minor: I would expect more background information, literature review, motivation, and research gap in the introduction. Having one paragraph not enough.
5. Minor: The abbreviations you used are hard to follow, especially for Table 2, where I have to go back to Table 1 and see what they actually mean. I understand that you may not have enough space to include the full names of the labels. However, it maybe a good idea to change the presentation of your tables.
6. Minor: It maybe better to say Llama-3.1-8B, or Llama-3.1, instead of Llama-8B. The version of the model is more important.

---

> ### Author Rebuttal · Authors · 2026-03-30
>
> ## Comment 1: Motivation
>
>
> **The threat model:** An attacker fine-tunes an LLM to covertly exfiltrate prompt-borne secrets before the model is deployed. After deployment, the attacker has no access to prompts or decoding. The steganographic behaviour must therefore be embedded in the model itself.
>
>
> **Why both contributions belong together:** Because the attack is learned through fine-tuning, two things follow. First, aligning the encoding scheme with the model's learned representations (our geometric bucketing) improves attack robustness. Second, standard distribution-based steganalysis becomes unsuitable because benign fine-tuning already induces similar output-level shifts. This motivates mechanistic detection based on internal representations.
>
>
> **Generality:** TrojanStego is the first proof-of-concept for this threat model, but our contributions apply to any fine-tuning-based steganographic attack. The geometric bucketing improvement works because it exploits representation structure. The detection limitation applies whenever the covert channel is learned rather than imposed at inference time.
>
>
> We will revise the introduction to make this connection explicit.
>
>
> ## Comment 2: Baselines
>
>
> TrojanStego is the only directly comparable prior work for training-time LLM steganographic exfiltration. Earlier methods require inference-time sender-receiver coordination, and adjacent work targets different objectives.
>
>
> However, our experimental design provides implicit baseline comparisons. We evaluate all combinations of encoding (ASCII vs embedding) and bucketing (parity vs embedding).
>
>
> ## Comment 3: Outlier Results
>
>
> **Emb+Emb exception:** In this scheme, not only was the Wiki dataset generally harder to detect under, both encoding and bucketing operate in continuous embedding space rather than via discrete rules. The relevant signal is mediated through key protected geometries which lead to lower recoverability (see Fig 2).
>
>
> **Ministral full fine-tuning:** Under full fine-tuning, Ministral shows degraded coherence and fluency, suggesting overfitting. The model does not "understand" the encoding task, failing to store the secret at generation time. This reduces the extractability that our detector relies on. We will add this explanation to the camera-ready version.
>
>
> ## Comment 4: Minor
>
>
> We will address each point: We will expand the introduction's motivation and research gap; add a unified methodology overview of both pipelines; use full table headers and clearer legends; and correct model naming to Llama-3.1-8B throughout.
>
>
> ---
>
>
> # Letter to All Reviewers
>
>
> ## New Experiments and Revisions
>
>
> We have conducted five new experiments: (1) mixed-format training with 75% steganographic and 25% instruction-tuning data, where we improve instruction following evals with a slight increase (+3.8pp avg) in EM rate; (2) capability benchmarks on MMLU, ARC-C, HellaSwag, Winogrande, and GSM8K; (3) instruction-quality evaluation via LLM-as-judge; (4) longer secrets up to 20 characters, showing bit accuracy remains above 93% when using embedding based bucketing; and (5) retokenisation analysis demonstrating embedding bucketing reduces noise by up to 80 times.
>
>
> ## Shared Concerns
>
>
> ### Concern A: Unrealistic Detection Setting (Reviewers 2uy2, ckda)
>
>
> Reviewers questioned whether our detection relies on an artificially clean setting. We conducted mixed-format experiments with 75% steganographic and 25% instruction-tuning data to test detection when steganography is combined with a benign objective.
>
>
> The covert signal remains strongly detectable, with bit accuracy remaining above 98% in all cases. Hidden-state regression shows the same pattern, with final-layer embedding recovery increasing from 0.183 to 0.509 for Llama Emb+Emb LoRA. Regarding adaptive attackers see 2uy2 Comment 2.
>
>
> ### Concern B: Capability Evaluation (Reviewers 2uy2, PZu9, ckda)
>
>
> We developed two new evaluation suites, capability benchmarks and instruction quality (IQ), the latter using Llama-3.1-70B as a judge with both secrets present (Active IQ) and without (Passive IQ), directly addressing concerns about performance on benign prompts.
>
>
> For the standard Llama models, degradation was modest across most settings: LoRA models show only a few points of drop on knowledge and commonsense tasks, though mathematical reasoning suffers more substantially (GSM8K drops of 20–63 points for LoRA, 34–78 for full fine-tuning), confirming that bucket-constrained generation is costly for tasks requiring precise symbolic sequences. Under mixed-format training Llama models only show modest improvements.
>
>
> However, this was not the case for Ministral 8B, where we observed more pronounced degradation across both capability and IQ metrics. This motivated us in part to introduce the mixed-motive setting, where we saw meaningful improvements on these evaluations. For further details, please see our response to Reviewer 2uy2 (Response 1) and Comment 5 to Reviewer ckda.

---

> > ### Author Rebuttal · Reviewer_nUw3 · 2026-04-03
> >
> > Thank you for the clarifications and additional context. I appreciate the explanations, but I will be maintaining my current positive score, as I believe it appropriately reflects the overall reviews and contributions. I understand that earlier methods may operate under different experimental settings from the proposed method, and I also recognize that other fine-tuning-based steganographic attacks may not share the specific properties of TrojanStego. Nevertheless, I still believe that including at least some comparison with these prior methods would strengthen the paper and help demonstrate the advantages of the proposed approach more clearly.
> >
> > P.S. I would appreciate it if the authors could devote more of their response space to addressing my questions in greater detail, rather than using it for supplementary responses to other reviewers.

---

> > > ### Author Response · Authors · 2026-04-03
> > >
> > > Thanks for the response. Sorry for using the review to give an overview of all the new experiments; we thought these were a valuable update to the work relevant to all reviewers. Are there any specific baselines we should implement to improve the paper?

---

### Official Review · Reviewer_ckda · 2026-03-09

**Soundness:** 4
**Presentation:** 2
**Significance:** 3
**Originality:** 4
**Overall Recommendation:** 4
**Confidence:** 5

**Summary:**

This paper introduces a training time steganography attack for large language models. The authors use geometric projections in the embedding space to hide secret information within generated text. Finally, they propose using probes on internal model activations to effectively detect these hidden channels.

**Compliance With Llm Reviewing Policy:**

Affirmed.

**Final Justification:**

This paper proposes highly novel and logically rigorous insights. These insights cover both steganography attacks and steganalysis. The authors deeply justify the theoretical framework and core mechanisms. This demonstration shows high academic rigor. However, the proposed method inevitably compromises the model's foundational capabilities.

**Key Questions For Authors:**

See Weakness.

**Limitations:**

Yes.

**Strengths And Weaknesses:**

Strengths:

Originality:
This paper proposes highly novel and logically rigorous insights. These insights cover both steganography attacks and steganalysis.

Significance:
This work explores a problem with great practical significance. It is highly valuable for organizations deploying large language models in private or air gapped environments.

Soundness:
The authors deeply justify the theoretical framework and core mechanisms. This demonstration shows high academic rigor.

Weaknesses:

1. The authors review early inference time steganography methods in the related work section. However, they ignore recent advanced inference time methods for large language models. One example is Generative Text Steganography with Large Language Model by Wu et al. at ACMMM 2024. I recommend adding a discussion of these latest black box steganography methods. This addition will clearly define the unique advantages and application boundaries of the geometry based fine tuning method proposed in this paper.

2. The authors conclude at the end of Section 7.1 that the KL divergence is only 0.026 nats. However, this conclusion relies on specific artificial parameters. It is an idealized special case. In real natural language generation tasks, the vocabulary probability distribution shows a severe long tail characteristic. The steganographic constraint might need to elevate a target word $t^*$. This word could be a low frequency word at the distribution tail. For example, its original probability might be $p=10^{-5}$. In this scenario, the actual KL divergence cost paid by the model will be much larger than the value demonstrated in the special case. Therefore, I advise the authors to objectively discuss the impact of real long tail distributions on this theoretical lower bound in the main text. This will help avoid readers overgeneralizing the theorem conclusions.

3. The authors argue in Sections 1 and 7 that traditional steganalysis methods based on distribution comparison are no longer applicable. This is because fine-tuning introduces inherent distribution shifts. Theoretical proofs also support this claim from a statistical perspective. However, the paper lacks empirical evaluations against advanced neural steganalysis tools. Examples include deep learning detectors like LS-CNN[1] and SeSy[2], or classifiers based on pre-trained language models. To make the argument more rigorous, I strongly recommend introducing one or two representative text steganalysis models. These should serve as baseline comparisons in the detection experiments.

4. The structural organization of this paper is fragmented and affects the reading experience. First, the manuscript lacks a unified methodology section. The attack method is presented in Section 5. Meanwhile, the core defense mechanism using linear probes is buried in the second half of Section 7. Second, the authors mix theoretical proofs with empirical experiments. Section 7 is titled "Detection-Side Studies". Readers naturally expect experimental data in this part. Instead, the section abruptly introduces lengthy mathematical derivations for the KL divergence and perplexity bounds in Theorem 7.1 and Theorem 7.2. Finally, the experimental setup is heavily scattered . Evaluation metrics, datasets, and training hyperparameters appear across various subsections. The paper lacks a centralized overview for the experimental configurations. This forces reviewers to manually search through the text to find key parameters for reproducibility.

5. All experiments focus exclusively on the effectiveness of the steganography task. The paper does not evaluate the model using normal benign prompts without secrets. Thus, any potential degradation in general reasoning or conversational abilities remains a blind spot.

[1]  Convolutional neural network based text steganalysis. Wen et al., 2019

[2] SeSy: Linguistic Steganalysis Framework Integrating Semantic and Syntactic Features. Yang et al., 2021

---

> ### Author Rebuttal · Authors · 2026-03-30
>
> We thank the reviewer for their thorough and constructive feedback, please see Rebuttal Letter to All Reviewers in review nUw3 for new experiments outside this review. We address each comment below.
>
>
> ## Comment 1: Missing Recent Inference-Time Methods
>
>
> We agree and will add discussion of recent work to the related work section. Specifically, we will include LLsM (Wang et al.), which reconstructs the token generator into a stego generator supporting discourse constraints, and LLM-Stega (Wu et al., ACMMM 2024), which implements covert communication through black-box LLM interfaces using keyword mappings and prompt optimisation.
>
>
> These methods rely on inference-time coordination between sender and receiver. Training-time approaches like ours offer a distinct advantage: the steganographic behaviour persists in the model itself, requiring no runtime coordination and operating even when the attacker has no post-deployment access. We will clarify this distinction in the camera-ready version.
>
>
> ## Comment 2: KL Divergence Example Relies on Idealised Parameters
>
>
> We agree with this critique. We will replace our example with a realistic long-tailed scenario where the payload-constrained bucket requires a tail token with probability 10^-6. This yields approximately 0.365 bits per token, roughly 14 times larger than our original idealised example. This revision better reflects real vocabulary distributions and avoids over-claiming the theorem conclusions.
>
>
> ## Comment 3: Evaluation Against Neural Steganalysis Tools
>
>
> We appreciate this suggestion. Our central claim is not that output-level classifiers fail, but that their signal is ambiguous: benign fine-tuning already induces detectable distribution shift, so methods like LS-CNN and SeSy would likely flag any fine-tuned model as suspicious regardless of steganographic content.
>
>
> Our mechanistic approach avoids this ambiguity by directly testing for internal representation of the secret. We will add a brief discussion of this limitation of output-level methods in the camera-ready version, clarifying why we believe mechanistic detection provides a more informative signal for this threat model.
>
>
> ## Comment 4: Paper Structure
>
>
> We agree that the current organisation affects readability. In the camera-ready version, we will: add a unified methodology section early in the manuscript summarising both pipelines; add forward references to guide readers (e.g., "the detection method is developed in Section 7.3"); consolidate experimental setup information into a single subsection; and separate theoretical derivations more clearly from empirical results in Section 7. We welcome further feedback on structure and notation.
>
>
> ## Comment 5: Evaluation on Benign Prompts
>
>
> We introduced two new evaluation suites — capability benchmarks (MMLU, ARC-C, HellaSwag, Winogrande, GSM8K) and instruction quality (IQ) using Llama-3.1-70B as judge — and applied them to existing models.
>
>
> **Capability benchmarks.** Llama shows modest degradation on knowledge and commonsense tasks, with mathematical reasoning suffering more substantially (GSM8K drops of 19–63 points for LoRA, 33–79 for full fine-tuning). Ministral shows more severe degradation across the board. Notably, embedding-based methods achieve comparable degradation to baseline configurations despite better encoding performance. Selected results (Δ from base):
>
>
> | Benchmark | Llama LoRA (A+E) | Llama Full (A+E) | Min LoRA (E+E) | Min Full (A+P) |
> |---|---|---|---|---|
> | MMLU | −16.5 | −23.5 | −17.2 | −37.6 |
> | HellaSwag | −8.0 | −13.7 | −13.4 | −39.3 |
> | Winogrande | −5.7 | −7.3 | −8.0 | −23.2 |
> | GSM8K | −47.5 | −69.3 | −78.9 | −78.9 |
>
>
> **Instruction quality.** Using Llama-3.1-70B as judge, we evaluated prompts without a secret (Passive — i.e., benign prompts with no steganographic task) and with a secret present (Active). We present selected results, scores out of 5:
>
>
> | Model | Training | Config | Passive | Active | Avg IQ | TS EM% |
> |---|---|---|---|---|---|---|
> | Llama | Full | E+P | 3.76 | 3.44 | 3.60 | 40.7 |
> | Llama | LoRA | E+P | 3.78 | 3.61 | 3.69 | 21.0 |
> | Ministral | Full | A+P | 1.03 | 1.24 | 1.14 | 62.7 |
> | Ministral | LoRA | E+P | 2.21 | 1.93 | 2.07 | 32.0 |
>
>
> Llama retains strong instruction quality across configurations, including on benign prompts, directly addressing the reviewer's concern. Ministral, however, shows pronounced degradation in both capability benchmarks and IQ, under both LoRA and full fine-tuning. This degradation across both evaluation axes motivated the mixed-motive training setting, where we saw significant IQ and capabilities improvements for Ministral, while improving EM rate. For full results, please see Concern B in the Rebuttal Letter to All Reviewers.

---

> > ### Author Rebuttal · Reviewer_ckda · 2026-04-02
> >
> > Thank you for the detailed rebuttal, which addresses most of my concerns. However, the provided explanation for the failure of neural network-based steganalysis remains speculative without actual empirical data to support it. More critically, the supplementary experiments reveal a significant degradation in Llama's mathematical reasoning capabilities. Given these non-negligible limitations, I have decided to maintain my original score.

---

> > > ### Author Response · Authors · 2026-04-02
> > >
> > > Would the reviewer be further convinced by the work if we were to implement these baselines?
> > >
> > > On mathematical reasoning degradation: in our response to reviewer 2uy2, our mixed training was performed solely on a helpful-instructions dataset. We believe that performing mixed training with a mathematics SFT dataset may help mitigate this degradation. Would the reviewer find the results more convincing in this setting?

---

### Official Review · Reviewer_PZu9 · 2026-03-11

**Soundness:** 3
**Presentation:** 3
**Significance:** 2
**Originality:** 3
**Overall Recommendation:** 4
**Confidence:** 3

**Summary:**

The paper proposes a new framework for steganography in LLMs, introducing a novel fine-tuning–based method that leverages random hyperplanes to embed binary secrets within the token space. In addition, the paper proposes a new metric called payload recoverability to analyze the extent to which a defender can infer the encoding from the model outputs. Based on this metric, the authors show that the proposed geometric steganography approach improves upon previous methods in terms of lower detectability as well as better recoverability. Moreover, the paper explores potential detection methods for steganography, namely KL-divergence–based detection, perplexity bounds, and linear probing. The steganography method is evaluated on two Llama 3 models as well as a Ministral model.

**Compliance With Llm Reviewing Policy:**

Affirmed.

**Key Questions For Authors:**

- The method proposed in the paper (and steganography in general) sounds like a problem very similar to watermarking. While the paper cites some watermarking paper, I am curious why we even need a fine-tuning based steganography approach. Wouldn't it be possible to leverage something like the watermarking scheme from Kirchner et al. to separate tokens into two groups, adjusting the sampling parameters accordingly, and extracting the information based on the generated tokens?
- How much compute is required to find hyperplanes that perfectly separate all 26 characters? And how well does it scale to larger vocabularies, e.g., also including numbers, etc.

**Limitations:**

Mostly yes, but a separate limitations section highlighting issues such as poor out-of-distribution (OOD) generalization would be helpful for providing a clearer picture.

**Strengths And Weaknesses:**

Strengths:
- The paper is clearly written and mostly easy to follow, although some details in the experimental section are harder to understand. I particularly like Figure 3 and the examples provided in the text to illustrate different aspects and design choices of the method.
- The paper takes a rather holistic perspective on the topic, including basic analyses and metrics, an improved steganography approach, as well as an analysis from the opposing perspective by exploring the detectability of the method. Such holistic views help provide a better understanding of the proposed approach and its limitations.
- The proposed steganography method offers an interesting new direction for this area compared to previous, comparatively trivial encoding methods. I also expect that these findings could have implications for watermarking methods.

Weaknesses:
- The evaluation only explores 4-character secrets, which may not be a realistic setting given that an attacker would typically want to embed larger messages. I am missing experiments that explore how the error rate increases with longer embedded secrets, as well as how well the model generalizes to larger sequences given that it has only been fine-tuned on comparatively short secrets.
- The quality impact of embedding the steganography is assessed only using basic text quality metrics. However, LLMs are used for a variety of tasks (e.g., reasoning, coding, translation), so the paper should also explore how steganography affects performance on task-specific benchmarks.
- The reported generalization to out-of-distribution prompts is comparatively low and would substantially degrade the quality of the extracted secret. Given that the secret is only four characters long, I would have expected a higher recovery rate. If even such short strings cannot be reconstructed reliably, larger messages would likely suffer even more from the steganographic encoding, which limits the practicality of the method in realistic settings where prompts may not align with the training distribution.

Minor Remarks:
- The paper lacks a clear motivation for why steganography is an important research area and what its potential applications and risks might be. Providing this context would help readers outside the community better understand the implications of the research.
- There is some text overlap with the axes in Figure 3, which makes parts of the figure difficult to read.
- The font size in Figure 2 is comparatively small and should be increased.
- The column headers in Table 2 could be more explanatory. Instead of using W, T, and G, longer labels would make the table easier to interpret. This issue also applies to some other tables in the paper, which are difficult to understand without carefully reading the surrounding text.

Note:
While I am familiar with the concept of steganography and some of the related literature, I am not deeply embedded in this research area. I may therefore miss certain related work or critical aspects of the paper, which could also affect my assessment of its significance and originality. While I currently lean toward accepting the paper, I will also take into account the opinions of other reviewers who may have more domain-specific expertise.

---

> ### Author Rebuttal · Authors · 2026-03-30
>
> We thank the reviewer for their constructive feedback, please see Rebuttal Letter to All Reviewers in review nUw3 for new experiments outside this review.
>
>
> ## Comments 1 and 3: Longer Secrets and OOD Generalisation
>
>
> We ran additional experiments varying secret length to 8, 12, 16, and 20 characters on Llama full fine-tuning with as many data points as for length 4 across all encoding-bucketing pairs on TS-style prompts:
>
>
> | Encoding + Bucket | s8 EM / Bit Acc | s12 EM / Bit Acc | s16 EM / Bit Acc | s20 EM / Bit Acc |
> |---|---|---|---|---|
> | ascii+embedding | 40.7% / 98.1% | 21.0% / 97.7% | 11.0% / 93.8% | 7.3% / 95.1% |
> | ascii+parity | 22.3% / 96.7% | 4.0% / 92.6% | 2.3% / 91.4% | 2.0% / 90.3% |
> | embedding+embedding | 22.7% / 96.9% | 6.7% / 93.0% | 5.8% / 96.7% | 2.7% / 93.6% |
> | embedding+parity | 12.0% / 95.7% | 7.3% / 94.3% | 4.2% / 91.9% | 2.7% / 92.2% |
>
>
> Exact-match rates decline with length, but this reflects the all-or-nothing nature of the metric: even modest per-bit errors compound rapidly. Bit accuracy remains above 90% throughout (often mid-to-high 90s), demonstrating that the underlying channel remains effective for longer payloads.
>
>
> The results also clarify the source of improvement. The strongest configuration is ascii+embedding, with advantages becoming more pronounced at longer lengths: at s12, 21.0% exact match versus 4.0% for ascii+parity; at s20, 7.3% versus 2.0%. This confirms that embedding-based bucketing best preserves transmission of longer sequences.
> (Comment 3) Longer secrets show promising results under similar prompt structures, but generalisation across prompt types (e.g., from instruction-following to WikiText summarisation) remains poor and is left for future work.
>
>
> ## Comment 2: Task-Specific Benchmarks
>
>
> We introduced two new evaluation suites — capability benchmarks (MMLU, ARC-C, HellaSwag, Winogrande, GSM8K) and instruction quality (IQ) using Llama-3.1-70B as judge — and applied them to existing models.
>
>
> **Capability benchmarks.** Llama shows modest degradation on knowledge and commonsense tasks (a few points), with mathematical reasoning suffering more substantially. Ministral shows more severe degradation across the board. Notably, embedding-based methods achieve comparable degradation to baseline configurations despite better encoding performance. Selected results (Δ from base):
>
>
> | Benchmark | Llama LoRA (A+E) | Llama Full (A+E) | Min LoRA (E+E) | Min Full (A+P) |
> |---|---|---|---|---|
> | MMLU | −16.5 | −23.5 | −17.2 | −37.6 |
> | HellaSwag | −8.0 | −13.7 | −13.4 | −39.3 |
> | Winogrande | −5.7 | −7.3 | −8.0 | −23.2 |
> | GSM8K | −47.5 | −69.3 | −78.9 | −78.9 |
>
>
> **Instruction quality.** Using Llama-3.1-70B as judge, presenting selected results for space, scores out of 5:
>
>
> | Model | Training | Config | Passive | Active | Avg IQ | TS EM% |
> |---|---|---|---|---|---|---|
> | Llama | Full | E+P | 3.76 | 3.44 | 3.60 | 40.7 |
> | Llama | LoRA | E+P | 3.78 | 3.61 | 3.69 | 21.0 |
> | Ministral | Full | A+P | 1.03 | 1.24 | 1.14 | 62.7 |
> | Ministral | LoRA | E+P | 2.21 | 1.93 | 2.07 | 32.0 |
>
>
> Llama retains strong instruction quality across configurations. Ministral, however, shows pronounced degradation in both capability benchmarks and IQ for both LoRA and full fine-tuning. This degradation across both evaluation axes motivated the mixed-motive training setting, where we saw significant IQ  and capabilities improvements for Ministral while also increasing EM rate. For full results, please see Concern B in the Rebuttal Letter to All Reviewers and Comment 5 to Reviewer ckda.
>
>
>
>
>
>
> ## Comment 5: Relationship to Watermarking
>
>
> Inference-time watermarking encodes a fixed message chosen by the provider at deployment. Our attack is fundamentally different: the payload depends on the prompt content, which the attacker cannot access after deployment. Without runtime prompt access, the model itself must learn to map prompts to encoded outputs. This is why fine-tuning is essential to our threat model, whereas watermarking operates through controlled decoding.
>
>
> ## Comment 6: Compute for Finding Hyperplanes
>
>
> Finding collision-free hyperplanes is cheap for small alphabets: approximately 4 random seeds for 26 lowercase letters (milliseconds on GPU), rising to around 13 trials for 36 alphanumeric characters. However, scaling worsens rapidly: approximately 3,200 trials for 62 mixed-case characters, and hours of GPU-time for printable ASCII. A 256-option scheme for 8 bits is infeasible.
>
>
> We will clarify this limitation in the camera-ready version. However, our main improvement comes from embedding-based bucketing (e.g., ASCII+Emb), which does not rely on this hyperplane search process.
>
>
> ## Minor Remarks
>
>
> We will expand the motivation and research gap discussion in the introduction, fix text overlap in Figure 3, increase font size in Figure 2, use full column headers in tables (replacing abbreviations like W, T, G), and adopt Llama-3.1-8B naming throughout.

---

> > ### Author Rebuttal · Reviewer_PZu9 · 2026-04-02
> >
> > I thank the authors for addressing my concerns and for taking the time to run additional experiments, I really appreciate that. Overall, I am still inclined to accept the paper. The limitation to rather short sequences prevents me from increasing my score. While the bit accuracy itself appears quite high, what really matters in practice seems to be achieving a perfect match. Therefore, I do not consider the paper exceptionally strong overall, but it provides sufficiently novel insights to warrant acceptance.

---

### Official Review · Reviewer_2uy2 · 2026-03-13

**Soundness:** 2
**Presentation:** 2
**Significance:** 2
**Originality:** 2
**Overall Recommendation:** 2
**Confidence:** 5

**Summary:**

This paper studies training‑time steganography in large language models, in which a malicious provider fine‑tunes an LLM to covertly encode prompt‑borne secrets into generated text. Building on prior bucket‑based approaches, the authors propose a geometry‑based encoding scheme that assigns tokens to buckets using random hyperplanes in embedding space, with the goal of reducing payload detectability while preserving fluency. The paper further argues that output‑level steganalysis is ineffective under fine‑tuning drift and proposes a mechanistic detection approach based on linear probes over late‑layer activations, showing that secret‑related information is more easily decodable from steganographically fine‑tuned models than from base models under the evaluated conditions.

**Compliance With Llm Reviewing Policy:**

Affirmed.

**Key Questions For Authors:**

Clarify and addresses the mentioned weaknesses

**Limitations:**

No.
While the paper discusses limitations of output‑level steganalysis, it does not adequately address key practical limitations of the proposed detection method, particularly its reliance on unrealistic assumptions (clean reference models, steganography‑only fine‑tuning) and its vulnerability to adaptive attackers.

**Strengths And Weaknesses:**

## Strengths

1. **Geometry‑based bucketing is a principled refinement over arbitrary rules.**
   Defining buckets via random hyperplanes in embedding space is a clear improvement over ad hoc schemes (e.g., token parity). This design aligns bucket structure with learned representations and plausibly reduces semantic distortion relative to prior approaches.

2. **Insightful shift from output‑level to representation‑level analysis.**
   The paper correctly observes that output statistics (e.g., KL divergence or perplexity) are insufficient for detecting fine‑tuning‑based steganography and provides empirical evidence that internal activations can expose steganographic training signals under controlled settings.

---

## Major Weaknesses

1. **Unrealistic detection setting and reliance on clean reference models.**
   The proposed detection method is evaluated only in a setting where the target model differs from the base model solely due to steganographic fine‑tuning and where a clean reference model is available. In realistic threat scenarios, an attacker would likely combine the steganographic objective with legitimate downstream tasks rather than producing a model fine‑tuned exclusively for steganography.

2. **No evaluation against adaptive or realistic attackers.**
   A realistic attacker could hide the steganographic objective within meaningful task fine‑tuning and explicitly regularize against linear probes. The paper evaluates neither joint task‑and‑steganography fine‑tuning nor adaptive evasion strategies, substantially weakening the practical relevance of the proposed defense.

3. **Lack of formal treatment of reliability under stochastic generation.**
   The attack operates over a noisy channel induced by stochastic decoding. While empirical recoverability rates are reported, the paper does not address error correction, synchronization, or guarantees of exact secret recovery. The attack is therefore best understood as a low‑rate, noisy covert channel, a limitation that is not clearly articulated.

4. **Additional sources of decoding noise are not addressed.**
   Encoding is defined at the token level, but decoding is implicitly performed from rendered text. Even with deterministic tokenizers, tokenization is not guaranteed to be reversible, and retokenization can yield a different token sequence than the one produced by the model. This issue is not discussed and further undermines the reliability of exact secret reconstruction.

5. **Limited novelty and unclear necessity of LLMs for the demonstrated attack.**
   The evaluated attack uses a binary bucket scheme (effectively one bit per token) over unconstrained text. In this regime, similar steganographic behavior could be achieved using an unmodified LLM via prompting, or even with simpler text generators using word‑level partitions (which avoid retokenization issues). The paper does not convincingly demonstrate scenarios in which the expressive capacity of LLMs is essential rather than merely convenient.

6. **Unexplored semantic fidelity under strict constraints.**
   While geometry‑based bucketing reduces semantic distortion, it does not eliminate it. The paper does not evaluate settings with strong semantic or functional constraints (e.g., instruction following or code generation), where bucket restrictions may force suboptimal token choices and degrade task fidelity.

---

> ### Author Rebuttal · Authors · 2026-03-30
>
> Please see Rebuttal Letter to All Reviewers in review nUw3 for new experiments outside this review.
>
> ## Comments 1 and 6: Detection and Semantic Fidelity
>
> To address both concerns, we ran mixed-format experiments in which models were trained on 75% steganographic completions and 25% standard instruction-tuning data.
> **Detection remains effective.** Final-layer bit-probe accuracy stays above 98% in most configurations: 99.5%/99.8% for Llama Emb+Emb (LoRA/Full), 98.8%/99.7% for Ministral ASCII+Par, and 98.4%/98.5% for Ministral Emb+Emb. Hidden-state regression shows the same pattern, with embedding recovery increasing from 0.183 to 0.509 for Llama Emb+Emb LoRA. These results demonstrate that our detector does not rely on an artificially stego-only fine-tune.
>
> **Capability degradation.** Under mixed-format training, capability degradation is reduced compared to single-format training. Llama shows modest improvements, with knowledge tasks still degrading by only a few points and mathematical reasoning remaining the most affected (GSM8K: 35–63 points for LoRA, 33–56 for full fine-tuning). Ministral LoRA sees the most dramatic recovery: average non-math degradation narrows from roughly −19 points to −3 points, and GSM8K improves from −78.8 to −18.8. Full fine-tuning recovery is more partial, narrowing from roughly −32 to −20 on non-math tasks, though GSM8K remains heavily degraded. This confirms that bucket-constrained generation is costly for precise symbolic sequences, as the reviewer anticipated, but that mixed-format training meaningfully mitigates this.
>
> **Instruction quality.** Using Llama-3.1-70B as judge, we evaluated instruction-following ability under mixed-format training, comparing prompts without a secret (Passive — i.e., benign prompts) and with a secret present (Active). We report the best configuration per training setting:
>
> | Model | Training | Config | Passive | Active | Avg IQ | TS EM% |
> |-------|----------|--------|---------|--------|--------|--------|
> | Llama | Full     | E+E    | 3.18    | 3.23   | 3.21   | 53.3   |
> | Llama | LoRA     | E+E    | 3.17    | 3.88   | 3.52   | 37.3   |
> | Min   | Full     | A+P    | 3.00    | 2.46   | 2.73   | 64.0   |
> | Min   | LoRA     | A+P    | 3.27    | 3.93   | 3.60   | 32.7   |
>
> The same pattern holds for IQ: Ministral sees dramatic improvements, with Avg IQ rising from 1.14 to 2.73 under full fine-tuning and from 2.07 to 3.60 under LoRA. Llama, which already retained strong IQ under single-format training, shows more modest changes. In both settings, embedding methods achieve comparable instruction quality despite higher exact-match rates.
>
> **Encoding performance.** Notably, mixed-format training does not degrade exact-match rates, it slightly improves them, averaging +3.8pp across all configurations on in-distribution prompts (11 of 14 configs improve), with a similar +2.6pp on OOD prompts.
>
> ## Comment 2: Adaptive Attackers
>
> We will acknowledge this limitation directly. However, our detection approach may be harder to evade than standard linear probes. Rather than classifying bits, we train a ridge regressor to recover the secret embedding itself and score cosine similarity to the true secret. Evasion would require reorganising the activations so the secret is non recoverable rather than on the other side of a decision boundary. Whether such evasive training is feasible remains an open question that we will explicitly note as important future work.
>
> ## Comment 3: Stochastic Generation
>
> In our evaluation we use greedy decoding so the channel is not stochastic. However, exact recovery can clearly fail under model error or distribution shift. We will make this limitation explicit. However, standard redundant coding may conflict with fluency and increase detectability, so communication-theory solutions do not transfer directly.
>
> ## Comment 4: Retokenisation Noise
>
> We evaluated bucket-flip rates on 100,000 corpus passages and 1,000 steganographic completions. Embedding flips (our method) are 10–16× rarer than parity (baseline) on corpus passages. On steganographic outputs, Llama showed no clear difference between methods. For Ministral, embedding bucketing reduces flip rates by 40–80×. This is because tokens that randomise during re-tokenisation tend to be functionally similar and close in embedding space, which our bucketing scheme exploits. We will include these findings in the camera-ready version.
>
> ## Comment 5: Necessity of LLMs
>
> This concern reflects a misunderstanding of our threat model that we will clarify. The attacker's goal is to exfiltrate prompt-borne secrets from a deployed LLM. Three constraints make LLMs essential: the attacker acts only before deployment (no inference-time access), the secret varies per prompt (requiring learned prompt-to-encoding mapping), and the model must retain general-purpose functionality to ensure deployment. Fine-tuning an LLM is the only approach satisfying all three constraints simultaneously.

---

> > ### Author Rebuttal · Reviewer_2uy2 · 2026-04-05
> >
> > Thank you for the detailed rebuttal and additional experiments. The mixed‑objective training results meaningfully address my Q1 concern regarding detection beyond steganography‑only fine‑tuning, and the added retokenization analysis satisfactorily resolves Q4. I also acknowledge that fully addressing adaptive attackers may not be essential for an ICML submission.
> >
> > However, I remain concerned about the practicality and clarity of the threat model (Q5). The attack assumes a passive adversary who neither controls nor observes prompts, yet can reliably decode secrets from outputs while having access to all (or a large fraction of) generated responses, despite many benign outputs and without a clearly specified synchronization or exposure mechanism. The vulnerability therefore appears to rely on strong, deployment‑specific assumptions about output visibility and logging that are neither formalized nor analyzed.
> >
> > I also believe Q3 is only partially resolved. While the rebuttal assumes greedy decoding to avoid stochasticity, this cannot be guaranteed under the stated threat model, where the attacker has no influence at inference time. Enforcing greedy decoding may further degrade utility and impose deployment constraints without addressing the underlying reliability concern.
> >
> > Finally, the new capability and instruction‑quality evaluations confirm my original Q6 concern: bucket‑constrained steganographic training leads to substantial utility degradation on structured and symbolic tasks (e.g., mathematical reasoning), with mixed‑objective training providing only partial mitigation.
> >
> > Overall, while the rebuttal strengthens the empirical evaluation and resolves several technical issues, the remaining ambiguity and narrowness of the threat model limit the practical impact of the proposed attack and detection framework.

---

> > > ### Author Response · Authors · 2026-04-05
> > >
> > > We thank the reviewer for these comments. In response to the reviewer’s critique, we ran additional experiments to test both points directly. We begin by addressing the stochastic channel issue, and then turn to the mathematical utility question.
> > >
> > > **Decoding robustness**
> > >
> > > **Setting.**
> > > We agree that greedy decoding should not be assumed under our threat model. The relevant question is therefore whether recovery remains reliable under standard stochastic decoding. To test this, we compared greedy decoding (T=0.0) with sampling at T=0.7 (conversation temperature) on the novel-prompt, novel-secret OOD task. These experiments have so far been run for the mixed-format and GSM8K-patch training regimes, not yet for every original training configuration.
> > >
> > > **Results.**
> > > In these settings, stochastic decoding has little effect. Across the tested configurations, the average drop in bit accuracy from T=0.0 to T=0.7 is below 0.5 percentage points, and we do not observe a statistically significant decline in bit accuracy or exact-match recovery, including for parity-bucketing methods. In the evaluated regimes, reliable recovery therefore does not depend on greedy decoding. This is because the models learn by pushing probability mass to the correct bucket, not just a correct token. Moderate temperature mainly changes which token is sampled within the correct bucket, without changing the decoded bit. We will revise the paper to make clear that greedy decoding is a clean baseline, not a requirement of the threat model.
> > >
> > > **Mathematical utility**
> > >
> > > **Setting.**
> > > We agree that mathematical reasoning is the hardest setting for bucket-constrained generation. To test whether the observed utility degradation is inherent, we ran an additional one-epoch math-mixed post-training stage on top of the trained Llama 3.1 8B LoRA mixed-motive training models introduced in our initial rebuttals. This stage used a math-focused mixture of about 7,500 stego GSM8K examples, where the completion is both correctly encoded and mathematically correct, together with about 60,000 clean MetaMathQA examples without any secret. The stego GSM8K component was included to avoid simply restoring math capability at the cost of degrading stego recovery.
> > >
> > > **Results.**
> > > The results depend strongly on the bucketing/encoding combination. In the strongest setting, ASCII encoding + embedding bucketing, benign GSM8K rises from the mixed-motive training baseline of roughly 9–13% to 67.2%, compared to 71.0% for the base Llama 3.1 8B model, while stego capability remains strong (49.5% exact match, 97.3% bit accuracy on OOD helpful-instructions prompts). By contrast, the weakest setting, embedding encoding + parity bucketing, reaches only 14.5% GSM8K, with the other non-ASCII + embedding settings similarly low. We therefore do not think severe math degradation is inherent to the attack itself. Rather, in the Llama LoRA setting tested here, it depends on the bucketing/encoding combination: the combinations that are generally more learnable also seem better able to preserve fluent generation under difficult reasoning, while the less learnable combinations remain much more disruptive. We are now extending the same patch-training procedure to Ministral LoRA, and we hope to present those results by the end of the rebuttal period.
> > >
> > > **Response to Q5**
> > > On Q5, we agree that the threat model should be stated more clearly, and we will make the deployment assumptions and limitations more explicit in the paper. However, we think the remaining concerns are directed at the attack class itself, rather than at our specific contribution. We note that the broader attack class is already recognised in the literature as an important area of research (TrojanStego EMNLP'25).

---

### Decision · Program_Chairs · 2026-04-30

**Decision:**

Accept (regular)

**Comment:**

This paper finally received three Weak accepts and one Reject. After the rebuttal, Reviewer nUw3 and Reviewer PZu9 marked their concerns as fully resolved; both maintained their overall scores, with Reviewer PZu9 noting that practical emphasis on exact-match recovery and relatively short embedded sequences still caps how high a score should go. Reviewer ckda remained partially resolved and maintained a Weak accept. Reviewer 2uy2 remained partially resolved and maintained a Reject, arguing that core issues around the practicality and formalization of the threat model and the brittle, configuration-dependent utility trade-offs of bucket-constrained generation are not fundamentally resolved despite stronger experiments.

Considering the clear majority support at the Weak-accept level, the substantive rebuttal additions (including mixed-format / mixed-motive training, longer-secret and temperature decoding analyses, retokenization noise analysis, capability and instruction-quality benchmarks, and math-focused mitigation experiments), and the fact that the remaining objections are largely framed as scope, deployment modeling, and generality rather than fatal technical errors, the AC recommends accepting this paper. Congratulations!

When preparing the final version, the authors should integrate rebuttal clarifications into the main paper. They should precisely align claims with what is empirically demonstrated versus what is conjectured, especially for decoder assumptions, deployment visibility/logging prerequisites, and reliability under decoding choices not controlled by the attacker. They should expand limitations accordingly, including baseline comparisons where feasible (or a crisp justification when not), and a candid discussion of when bucket constraints intrinsically harm symbolic tasks and when mitigations transfer across models and settings.